# *Streptococcus pneumoniae* synchronizes the states of cell wall peptidoglycan acetylation and genome methylation by programmed DNA inversions

Xiu-Yuan Li[1☯], Ping He[1☯], Shaomeng Wang[1☯], Yusong Wang[2], Dingfei Yan[3], Xiaohui Liu[2], Haiteng Deng[3], Zhixing Feng[4], Juanjuan Wang[5]*, Jing-Ren Zhang[1]*

**1** Center for Infection Biology, School of Basic Medical Sciences, Tsinghua University, Beijing, China, **2** Metabolomics and Lipidomics Center, National Protein Science Facility, Tsinghua University, Beijing, China, **3** MOE Key Laboratory of Bioinformatics, School of Life Sciences, Tsinghua University, Beijing, China, **4** Department of Clinical Genetics, Xinhua Hospital affiliated to Shanghai Jiao Tong University School of Medicine, Shanghai, China, **5** National Key Laboratory of Veterinary Public Health and Safety, College of Veterinary Medicine, China Agricultural University, Beijing, China

☯ These authors contributed equally to this work.
* juanjuanwang@cau.edu.cn (JW); zhanglab@tsinghua.edu.cn (JRZ)

## Abstract

Bacterial cell wall peptidoglycan (PG) consists of alternating β-(1,4) linked *N*-acetylmuramic acid (NAM) and *N*-acetylglucosamine (NAG). The C-6 hydroxyl group of NAM is acetylated by transmembrane *O*-acetyltransferases post PG bio-synthesis in many pathogenic bacteria. This modification is important for bacterial resistance to lysozyme. It is also known that the extent of NAM *O*-acetylation varies greatly, depending on genetic background and growth phase. However, it remains unclear if the fluctuation of NAM *O*-acetylation has any function. In this study, we show that NAM *O*-acetylation functions as a potential extracellular signal of cellular metabolism for epigenetic response to nutrient conditions in human pathogen *Streptococcus pneumoniae* (pneumococcus). The *O*-acetylation was found to control reversible switch between opaque and transparent colony phases by modulating inversion reactions of DNA methyltransferase *hsdS* genes in the colony opacity determinant (*cod*) locus, and thereby phase-defining genome methylation pattern. The NAM *O*-acetylation made *S. pneumoniae* adopt the HsdS_{A1} methylome and opaque colony phase, whereas the lack of this modification favored the HsdS_{A3} methylome and transparent colony phenotype. Further analysis revealed that the major autolysin LytA and multiple other proteins are required for the *O*-acetylation-dependent control of epigenetic machinery. Lastly, the extent of NAM *O*-acetylation was found to correlate with the cellular level of the acetyl donor acetyl-CoA and glucose. These data support the postulation that *S. pneumoniae* uses NAM *O*-acetylation as an extracellular marker of cellular acetyl-CoA to synchronize nutrient availability with bacterial lifestyle by epigenetic modulation of cellular metabolism.

**Data availability statement:** All the data involved in this study are available in the manuscript, supplementary figures and tables. The genome sequencing data of TH11857 are available in the NCBI database under the accession of SRR29649535. The raw data of SMRT sequencing in this study have been deposited in NCBI under the following accessions: SRR29286272 (ST606), SRR29286270 (TH13742), SRR29286271 (TH14720) and SRR32454972 (TH13740). The RNA-seq data are available in the NCBI database under the following accessions: SRR32455537 (ST606), SRR32455536 (TH14720) and SRR32455535 (TH16152).

**Funding:** This work was supported by grants from the National Key Research and Development Program of China (No. 2023YFc2306301 to J-.R.Z), the National Natural Science Foundation of China (No. 82330071 to J-.R.Z and No. 32100141 to J.W), and the Tsinghua Initiative Scientific Research Program (No. 20243080033 to J-.R.Z). The funders had no role in study design, data collection and analysis, decision to publish, or preparation of the manuscript.

**Competing interests:** The authors have declared that no competing interests exist.

## Author summary

Bacterial cell wall peptidoglycan (PG) consists of two basic sugars: *N*-acetylmuramic acid (NAM) and *N*-acetylglucosamine (NAG). The C-6 hydroxyl group of NAM is acetylated in many pathogenic bacteria after the PG is synthesized, which is important for bacterial resistance to lysozyme. The extent of NAM *O*-acetylation varies greatly, depending on genetic background and growth phase. However, it is unknown whether the fluctuation of NAM *O*-acetylation has any role in bacterial biology. Here, we show that human pathogen *Streptococcus pneumoniae* uses the level of NAM *O*-acetylation as an extracellular signal to epigenetically regulate cellular metabolism and phase variation in colony opacity. This is accomplished by *O*-acetylation-dependent modulation of inversion reactions of DNA methyltransferase *hsdS* genes in the colony opacity determinant (*cod*) locus, which leads to reversible switch among multiple methylation patterns of bacterial genome. Several extracellular, transmembrane and intracellular proteins are found to constitute a signaling circuit to connect the NAM *O*-acetylation and *hsdS* gene inversions. This work has thus uncovered a previously uncharacterized epigenetic mechanism of bacterial adaptation to nutrient availability.

## Introduction

*Streptococcus pneumoniae* (pneumococcus) is a natural colonizer of the human nasopharynx, but also causes invasive infections in normally sterile host niches, such as the lung (pneumonia), blood (bacteremia), and brain (meningitis) [1]. The bacterium encounters diverse host conditions with fluctuations in the availability and type of nutrients. While nutrients are scarce at the mucosal surfaces of the upper airway as a mechanism of host immunity, they are generally more available in the lung, blood, and brain [2,3]. However, it is largely unknown how *S. pneumoniae* metabolically adapts to the drastic differences in the nutrient supply between starvation and feast conditions during the colonization and invasive infection, respectively.

*S. pneumoniae* adopts two morphological states as manifested by the transparent (T) and opaque (O) colonies on transparent agar plates [4,5]. The T variant produces a relatively thinner polysaccharide capsule, and thereby is more capable of epithelial adhesion [4,6–8]. In contrast, the O counterpart possesses a thicker capsule with a stronger capacity of evading opsonophagocytic killing. These *in vitro* phenotypic differences are associated with the pneumococcal behaviors in animal models. The T variant is more prevalent in nasopharyngeal colonization, whereas pneumococci from the bloodstream mostly form O colonies [4,6]. It has become increasingly apparent that the T and O variants of *S. pneumoniae* represent two distinct cellular phases of nutritional adaptation. In particular, the T form adopts a "frugal" state whereas the O variant takes on a "luxurious" mode [5].

Pneumococcal cell wall consists of the lateral layers of peptidoglycan (PG) and vertical chains of cell wall teichoic acids (WTAs) (see Fig 1A) [9]. PG is composed of alternating β-(1,4) linked *N*-acetylmuramic acid (NAM) and *N*-acetylglucosamine

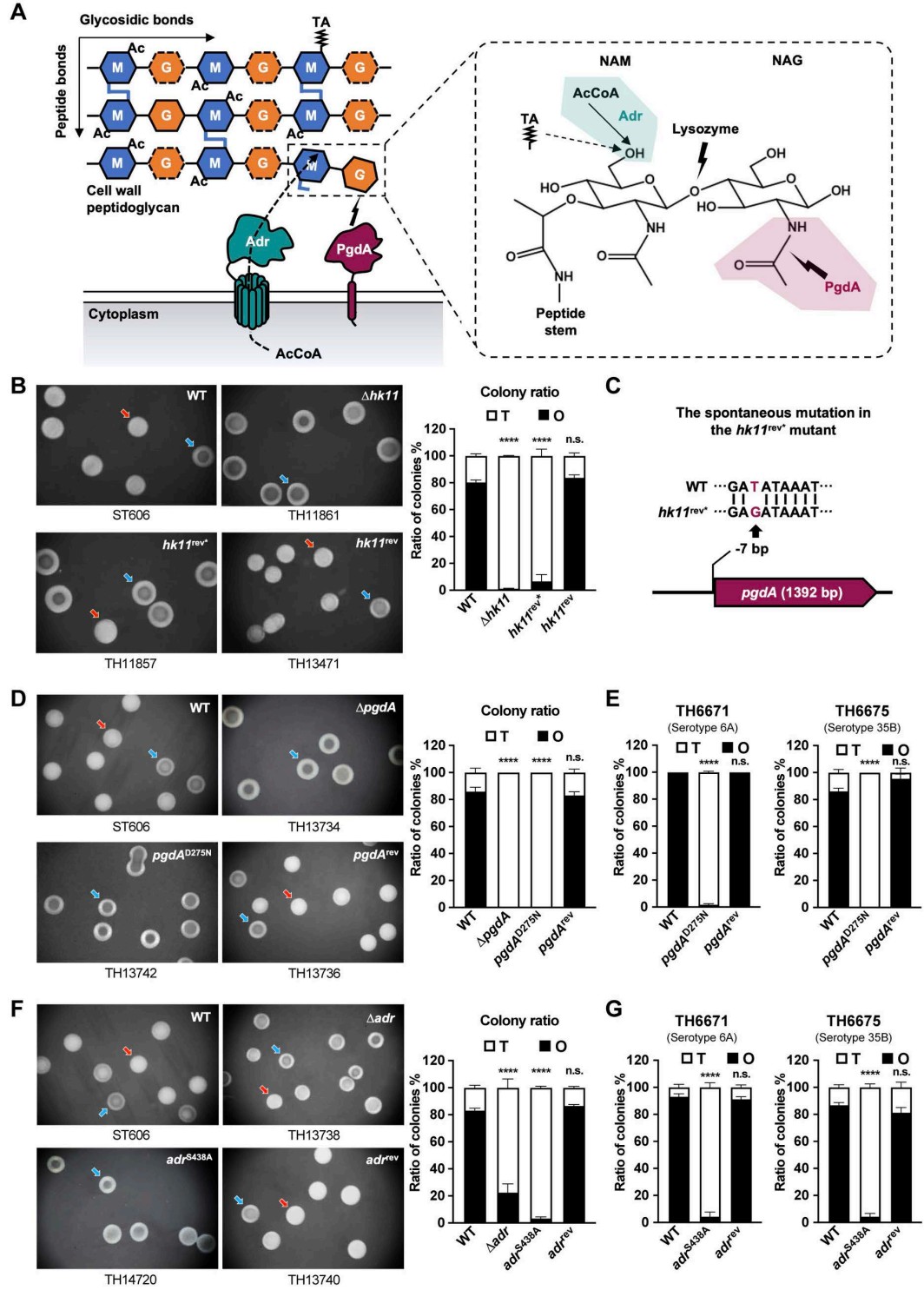

**Fig 1. Functional impact of PG modifications on colony opacity of *S. pneumoniae*. (A)** Illustrative depiction of the two post-synthetic PG modifications. M, *N*-acetylmuramic acid (NAM). G with the dash line, glucosamine. NAG, *N*-acetylglucosamine. AcCoA, acetyl-CoA. TA with the polygonal line, teichoic acid. **(B)** Colony phenotypes of ST606 (serotype 19F, WT) and isogenic *hk11* mutants on catalase-supplemented TSA plates. Representative opaque (O, red) and transparent (T, blue) colonies are presented (left panel) and quantified (right panel) from three replicates. **(C)** Illustration of the

mutation upstream of the *pgdA* coding region in ST606 *hk11*^rev* (TH11857). **(D)** Colony phenotypes of ST606 (WT) and isogenic *pgdA* mutants are presented as in **(B)**. **(E)** O and T colony ratio of *pgdA* mutants in serotypes 6A (TH6671) and 35B (TH6675) strains. **(F)** Colony phenotypes of ST606 (WT) and isogenic *adr* mutants. Data are presented as in **(B)**. **(G)** O and T colony ratio of *adr* mutants in serotypes 6A (TH6671) and 35B (TH6675) strains as in **(B)**.

(NAG). WTAs are made of choline-containing repeat units, and are covalently attached to the C-6 hydroxyl (C6-OH) group of the NAM residues [10]. The post-synthesis anchoring of WTAs to NAMs is necessary for growth phase-dependent and antibiotic-induced autolysis, which is catalyzed by the major autolysin LytA, after it is non-covalently attached to the choline residues of WTAs [11,12]. As observed in numerous bacterial species [13], *S. pneumoniae* employs the transmembrane *O*-acetyltransferase Adr to partially acetylate the C6-OH group of NAMs, which coincides with the anchorage locus of wall teichoic acid (WTA) within the cell wall [14]. NAM acetylation is required for pneumococcal resistance to lysozyme and antibiotic-induced autolysis [14–16]. This function agrees with the inhibitory effect of NAM *O*-acetylation on LytA binding to and cleavage of PG in *S. pneumoniae* [17]. However, excessive *O*-acetylation of NAM residues leads to bacterial growth arrest [18], which is consistent with the requirement of LytA for modification and growth of pneumococcal cell wall [9], and the septum positioning of the NAM *O*-acetylation enzymes [17,19]. The existing data show that the NAM *O*-acetylation is important for structural stability of PG and cell growth. The NAG residues are subjected to *N*-de-acetylation by the PgdA *N*-de-acetylase [20]. The *N*-de-acetylation enhances pneumococcal resistance to lysozyme [20].

The T and O variants of *S. pneumoniae* differ in their cell wall polysaccharides. The T form possesses more WTAs than the O counterpart [6,21–23]. The greater abundance of WTAs in the T variant is consistent with its relatively higher extent of autolysis. However, a causal relationship between pneumococcal PG modifications and phase variation has not been established. The previous studies have revealed that pneumococcal phase variation in colony opacity is epigenetically determined by reversible DNA variations in the DNA methyltransferase *hsdS* genes in the Spn556II/SpnD39III type I restriction-modification (R-M) system or the colony opacity determinant (*cod*) locus [24,25]. The *cod* locus consists of the *hsdR* (restriction endonuclease), *hsdM* (DNA methyltransferase, MTase), *psrA* (DNA invertase), and three homologous *hsdS* (*hsdS_A*, *hsdS_B* and *hsdS_C*) genes [25,26] (see Fig 2A). As the sequence recognition subunit of type-I RM DNA methyltransferase, *hsdS_A* encodes two target recognition domains, each of which recognizes half of the type-I RM methylation bipartite sequence, while *hsdS_B* and *hsdS_C* are not expressed. Instead, *hsdS_B* and *hsdS_C* serve as the templates for PsrA-catalyzed DNA inversions, which generate six *hsdS_A* allelic variants (*hsdS_{A1}* to *hsdS_{A6}*) [25,26]. Each HsdS_A variant recognizes a unique methylation sequence in pneumococcal genome and thus forms a distinct genome methylation pattern or methylome [25,27]. Among the six *hsdS_A* alleles (*hsdS_{A1-6}*) generated by *hsdS* inversions, only *hsdS_{A1}* makes pneumococci produce O colonies, whereas bacteria with the other five alleles form T colonies [25]. *hsdS_{A1}* and the other five alleles (*hsdS_{A2-6}*) are thus referred to as "opaque" and "transparent" alleles, respectively.

Our recent study has revealed that the phase-defining *hsdS* inversions are regulated by two-component regulatory systems [28] and a toxin-antitoxin system [29]. Genetic alterations in these systems significantly change the extent of *hsdS_{A1}*-carrying and thereby O variant in pneumococcal populations. These findings demonstrate that *S. pneumoniae* regulates *hsdS* inversions and genome methylation. However, the mechanisms of such the regulations remain largely undefined. In this study, we have characterized a serendipitous observation that post-biosynthesis modifications of pneumococcal cell wall control colony opacity. The deeper investigation found that the absence of NAM *O*-acetylation alters the *hsdS* inversion reactions in the *cod* locus, and thereby genome methylation pattern, which led to the discovery of multiple extracellular, transmembrane and cytoplasmic proteins as parts of the regulatory circuit. The biological implications of fluctuating NAM *O*-acetylation in bacterial adaptation to nutrient availability are discussed.

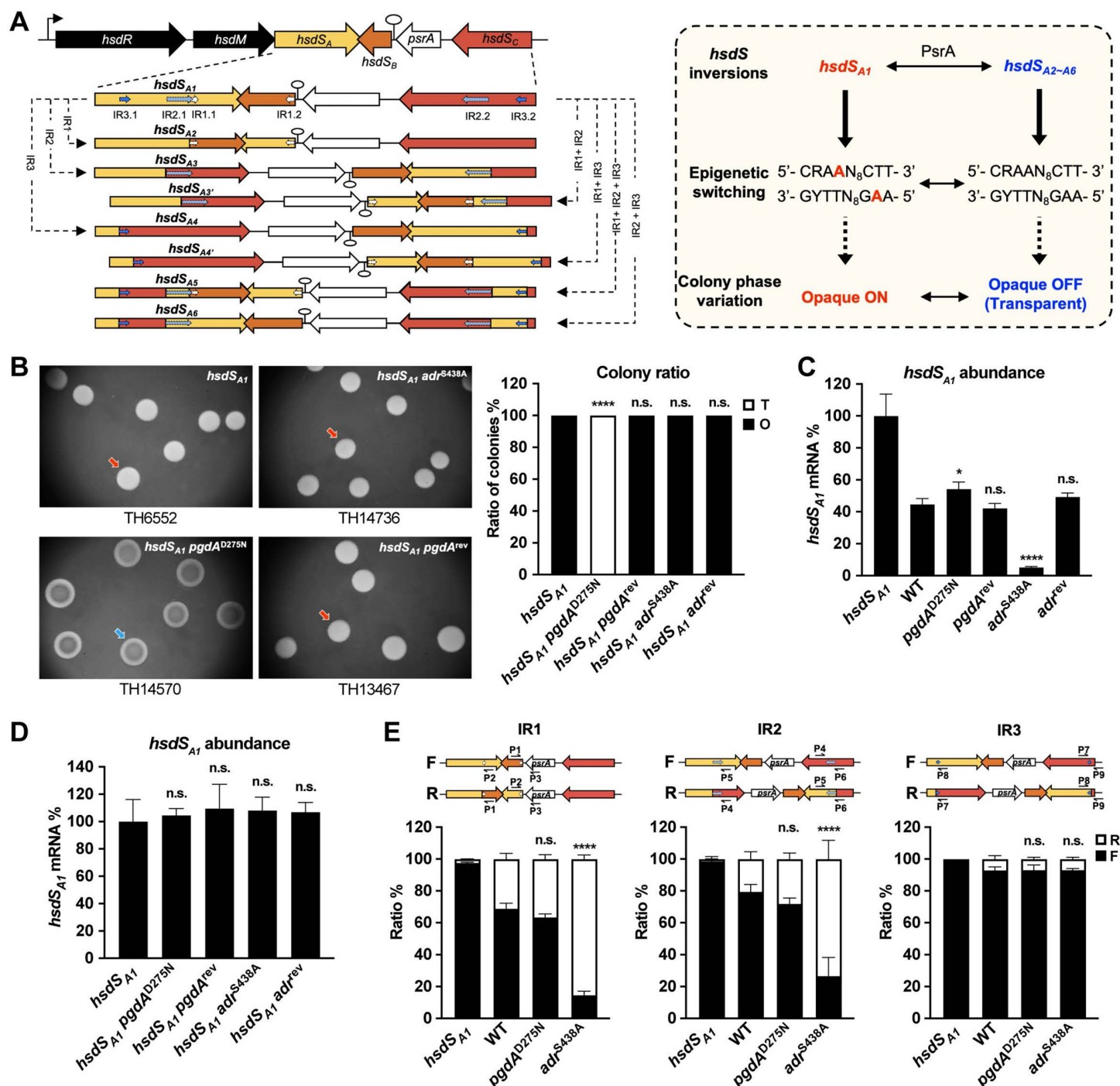

**Fig 2. Causal relationship between NAM *O*-acetylation and the orientation of *hsdS* inversions. (A)** Illustration of *hsdS* inversion-driven phase variation in colony opacity. The left panel depicts the coding regions of the type-I RM restriction enzyme subunit (*hsdR*) and methyltransferase sub-unit (*hsdM*), and the four genes in the *hsdS* inverton: *hsdS_A* (sequence recognition subunit), two non-expressing *hsdS_B* and *hsdS_C* genes, and *psrA* (invertase) in the *cod* locus. The right panel summarizes the relationship between allelic variants of *hsdS_A* and colony opacity. The methylated adenine nucleotides in the DNA motif by the HsdS_A1 MTase are highlighted in red. R = A or G, Y = T or **C. (B)** Colony phenotypes of *pgdA* and *adr* mutants in the *hsdS_A1*-fixed strain. Representative O and T colonies are presented as in [Fig 1B](). **(C)** Relative abundance of the *hsdS_A1* mRNA in *pgdA* (TH13742) and *adr* (TH14720) mutants. The *hsdS_A1* mRNA abundance of each strain was normalized by that of *hsdS_A1*-fixed mutant (*hsdS_A1*) and shown as mean ± s.d. of 3 replicates in a representative experiment. Significance between WT and mutants is presented. **(D)** Relative abundance of the *hsdS_A1* mRNA in

*pgdA* and *adr* mutants generated in the *hsdS*$_{A1}$-fixed background. Data represent mean ± s.d. for 3 replicates. **(E)** Ratio of IR1-, IR2-, and IR3-bound sequences in different orientations in *pgdA* and *adr* mutants. The DNA inversion mediated by each pair of IRs and the primers (P1 to P9) for detecting the ratio of each IR in different directions are illustrated at the top. F, forward. R, reverse. Bacterial ratio with different directions of IRs in each mutant is shown as mean ± s.d. of 3 repeats in a representative experiment. Significance between WT and mutants is presented.

## Results

### Peptidoglycan acetylation states determine the colony opacity phenotypes

Our previous study showed that the two-component system TCS11, consisting of the sensing kinase HK11 and response regulator RR11, promotes the HsdS$_{A1}$ genomic methylome and thereby the formation of O colony phase in *S. pneumoniae* [28]. While the parental strain (ST606) produced 80.3% O and 19.7% T colonies, the Δ*hk11* mutant (TH11861) showed only 1.1% O colonies (Fig 1B), which agrees with the importance of TCS11 in the formation of O colonies [28]. However, the Δ*hk11* colony phenotype remained the same when it was reverted with the intact *hk11* (*hk11*rev*, TH11857; Fig 1B). This result indicated that other gene(s) beyond *hk11* was involved in the colony phenotype. Whole genome sequencing of Δ*hk11*rev* (TH11857) revealed a single nucleotide replacement of the -7th thymine residue by a guanine residue in the 5' non-encoding region of the *pgdA* gene (*myy0814*) (Fig 1C). The *pgdA* encodes a cell wall NAG deacetylase (Fig 1A) [20]. Subsequent correction of this mutation (*hk11*rev, TH13471) restored the O/T colony ratio to that of WT (Fig 1B). Additional genetic manipulations of *pgdA* confirmed that the deacetylase activity of PgdA is required for O colony phase (Fig 1D). The role of PgdA in the phase variation was further verified in serotype-6A (TH6671) and serotype-35B (TH6675) strains (Figs 1E and S1).

Pneumococcal PG is also acetylated at its C-6 hydroxyl of NAM by the *O*-acetyltransferase Adr (Fig 1A) [14]. So, we next investigated the impact of NAM *O*-acetylation on pneumococcal colony opacity by targeted mutagenesis. As shown in Fig 1F, the proportion of O colonies in Δ*adr* mutant was reduced to 22.9% from 83.1% in WT. Revertant with the intact *adr* gene in the *adr* deletion background (*adr*rev) led to an increase in the O colony ratio to that of WT. To ascertain the role of Adr's enzymatic activity in pneumococcal colony opacity, we generated an *adr* mutant by replacing serine with alanine at position 438, a key amino acid for *O*-acetyltransferase activity [30]. Similar to the phenotype of the *adr* deletion mutant, the *adr*S438A mutant showed significant attenuation on the capacity of forming O colonies, generating only 3.4% O and 96.6% T colonies (Fig 1F). The causal relationship between Adr *O*-acetyltransferase activity and colony opacity was also confirmed in serotype-6A (TH6671) and serotype-35B (TH6675) strains. The *adr*S438A derivative of TH6671 produced a marginal level of O colonies (Figs 1G and S1). A similar degree of reduction in O colony was also observed in the *adr*S438A mutant of TH6675. The T-dominant phenotype in the *adr*S438A derivatives of TH6671 and TH6675 was reversed to that of parental strains with the intact *adr*. These findings demonstrated that Adr-catalyzed *O*-acetylation favors opaque colony phase.

### NAM *O*-acetylation stabilizes the O phase gene configuration

Previous studies have shown that the O and T phase variation is controlled by PsrA-catalyzed DNA inversions in the *cod* locus (Fig 2A) [25,27]. We thus tested whether the PgdA- and Adr-catalyzed PG modifications impact colony opacity through the programmed DNA inversions by generating *pgdA*D275N and *adr*S438A mutants in a pneumococcal strain that carried a fixed *hsdS*$_{A1}$ allele (TH6552) because the active site point mutation of the invertase PsrA uniformly formed O colonies [25]. Similar to the colony phenotype of *pgdA*D275N, the *hsdS*$_{A1}$-*pgdA*D275N mutant (TH14570) completely lost the ability to produce any O colonies. This phenotype was fully converted to that of parental strain when reverted with the intact *pgdA* (*hsdS*$_{A1}$-*pgdA*rev, TH13467) (Fig 2B). This result showed that PG *N*-deacetylation modulates pneumococcal colony opacity through a PsrA-independent mechanism. In sharp contrast, the absence of PsrA completely blocked Adr from affecting colony opacity. Compared with significant reduction of O colonies in the *adr*S438A strain, the *hsdS*$_{A1}$-*adr*S438A

strain (TH14736) showed uniform production of O colonies as the parental strain (Fig 2B). This functional dependence of Adr on PsrA suggested that NAM *O*-acetylation controls colony phase by modulating the *hsdS* inversions.

To ascertain the relationship between *hsdS* inversions and PG modifications in modulating colony phase, we assessed the ratio of *hsdS*$_{A1}$-genotype bacteria in the enzymatic inactivation *adr*$^{S438A}$ and *pgdA*$^{D275N}$ mutants by detecting the relative mRNA abundance of *hsdS*$_{A1}$ since only the HsdS$_{A1}$ methylation contributes to the formation of O colonies [28]. Normalized by the abundance of *hsdS*$_{A1}$ mRNA in the *hsdS*$_{A1}$-fixed mutant, a similar level of *hsdS*$_{A1}$ mRNA was detected in the WT and *pgdA*$^{D275N}$, while the *adr*$^{S438A}$ population exhibited a significant decrease in relative abundance of *hsdS*$_{A1}$ mRNA (Fig 2C). The reduced *hsdS*$_{A1}$ representation in the *adr*$^{S438A}$ strain was restored to the WT level with the intact *adr*. However, the impact of Adr on *hsdS* inversions became undetectable with the dysfunction of PsrA (Fig 2D). These data showed that NAM *O*-acetylation shapes pneumococci toward the *hsdS*$_{A1}$ allelic configuration in the *cod* locus.

PsrA catalyzes DNA inversions by recognizing three pairs of inverted repeats in the coding regions of *hsdS*$_A$ (IR1.1, IR2.1 and IR3.1), *hsdS*$_B$ (IR1.2), and *hsdS*$_C$ (IR2.2 and IR3.2) (Fig 2A) [25]. We characterized how NAM *O*-acetylation impacts *hsdS* inversions by detecting the forward and reverse orientation of the invertible sequences in the *cod* locus by quantitative PCR as described [26]. The *hsdS*$_{A1}$-fixed strain was used as a positive control to set the forward configurations for the IR1-, IR2- and IR3-bound sequences. Consistent with the O-dominant phenotype of WT strain, the IR1-, IR2-, and IR3-bound sequences were predominantly oriented in the forward direction (Fig 2E). While *pgdA*$^{D275N}$ mutant showed a similar genotype as WT, *adr*$^{S438A}$ mutant shifted the IR1- and IR2-bound sequences to the reverse direction.

We verified the impact of NAM *O*-acetylation on *hsdS* inversions in serotype-2 strain D39 and its *adr* derivative. As presented in S2 Fig, the *adr*$^{S438A}$ mutant of D39 showed significant reduction in the proportion of *hsdS*$_{A1}$-carrying variant (S2A Fig) and the orientation shift of IR-bound sequences (S2B Fig). These experiments indicated that NAM *O*-acetylation broadly modulates *hsdS* inversions in *S. pneumoniae*.

## NAM *O*-acetylation is required for the O phase methylome

Since the pneumococcal methylome catalyzed by the HsdS$_{A1}$ methyltransferase defines the O phase [26], we determined whether NAM *O*-acetylation affects the methylome by comparing DNA methylation status between WT and *adr*$^{S438A}$ strains. The 6-A methylation (6-mA) states in the sequences recognized by the six HsdS$_A$ allelic variants were determined by PacBio single molecule real-time (SMRT) sequencing (Fig 3A). Consistent with the phenotypic dominance of HsdS$_{A1}$-defined O phase in WT bacteria, nearly all the 2,060 loci of the HsdS$_{A1}$ motif (5'-CRA$^{m6}$AN$_8$CTT-3') were methylated in the WT genome, whereas less than 50% of the HsdS$_{A2}$ (5'-CRA$^{m6}$AN$_9$TTC-3') and HsdS$_{A3}$ (5'-CRA$^{m6}$AN$_8$CTG-3') counterparts were methylated (Fig 3B and 3C). This result indicated that the vast majority of bacterial cells in the WT population express the *hsdS*$_{A1}$ allele in the *cod* locus as reported previously [25,28]. In striking contrast, no methylation was detected for any of the HsdS$_{A1}$ motifs in the *adr*$^{S438A}$ genome, but the methylation rates for the HsdS$_{A2}$ and HsdS$_{A3}$ motifs were significantly increased. In particular, virtually all of the HsdS$_{A3}$ motifs (99.7%) became methylated in the *adr*$^{S438A}$ genome (Fig 3B and 3C). However, PacBio sequencing revealed the WT level of HsdS$_{A1}$ methylation in the *adr* revertant (Fig 3B and 3C), unequivocally demonstrating a causal relationship between NAM *O*-acetylation and O phase methylome. In contrast with the phenotypic disconnection between *hsdS* inversions and PgdA-driven colony opacity, the *pgdA*$^{D275N}$ strain showed loss of the HsdS$_{A1}$ methylome, as well as other HsdS$_A$ and even Spn556I/III methylomes (S1 Table), indicating the crucial role of NAG *N*-deacetylation in genomic methylation activity rather than modulating *hsdS* inversions. These striking differences in the methylome of *pgdA* mutant were not due to potential differences in sequencing depth, since there was a comparable level of total reads among the WT, *adr*$^{S438A}$ and *pgdA*$^{D275N}$ strains (S2 Table). Taken together, the NAM *O*-acetylation by Adr is absolutely required for the HsdS$_{A1}$ methylome in *S. pneumoniae*.

## LytA synchronizes NAM *O*-acetylation with the O phase methylome

The connection between NAM *O*-acetylation and *hsdS* inversions may be explained by two possibilities: 1) NAM *O*-acetylation enriches *hsdS*$_{A1}$-genotype bacteria due to growth advantage of the O variant, and 2) the status of NAM *O*-acetylation is

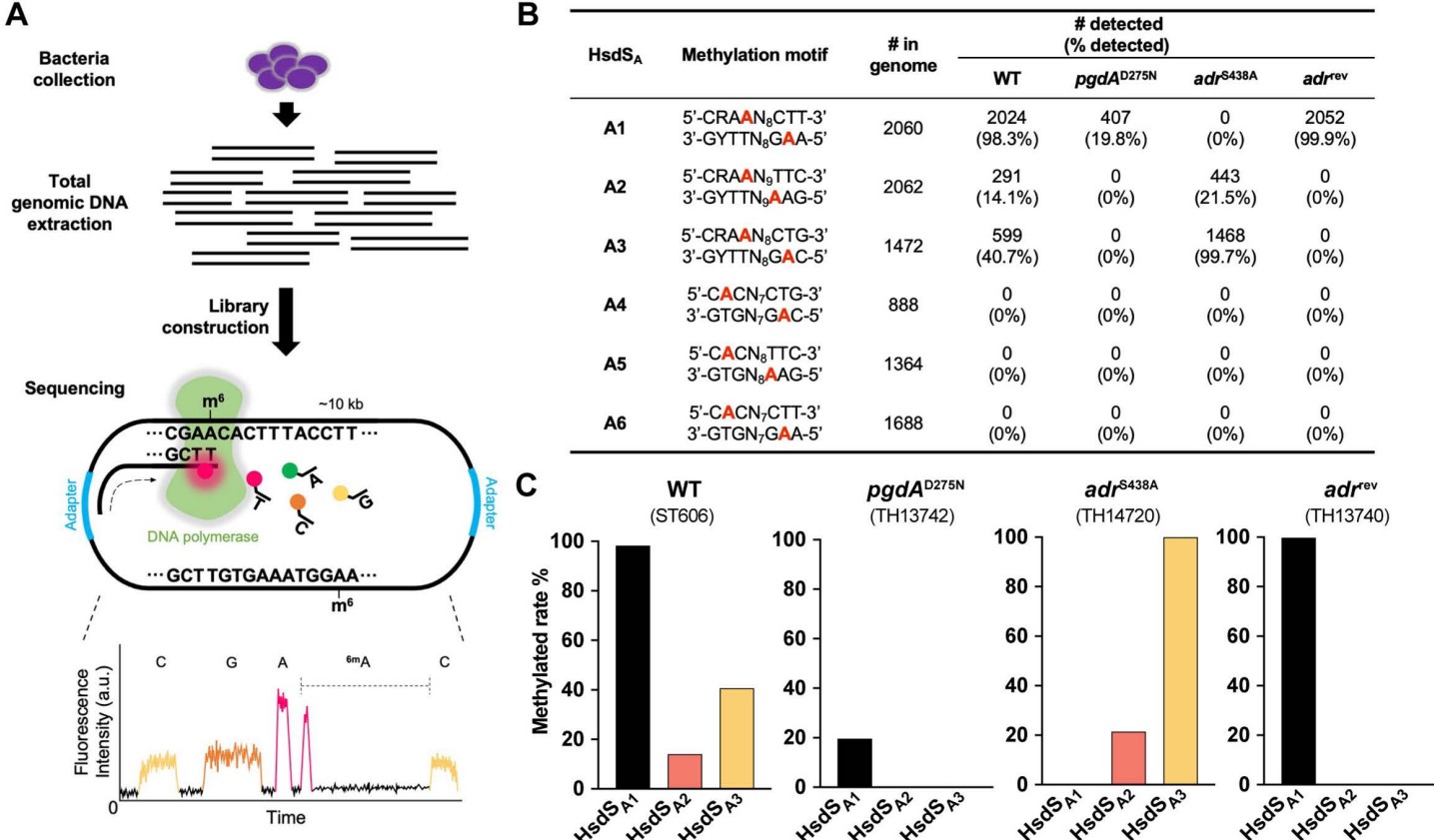

**Fig 3. The loss of HsdS_{A1} methylome in the absence of NAM *O*-acetylation. (A)** Illustration of experimental design for detecting pneumococcal genomic methylation by PacBio sequencing. **(B)** Genomic methylation by HsdS_{A1-6} MTases in WT, *pgdA*^D275N, *adr*^S438A and *adr*^rev strains. The methylated adenine nucleotides in each of six DNA motifs are highlighted in red. "# in genome" indicates the total copies of each methylation sequence in both strands of ST556 genome (accession CP003357.2). "% detected" and "# detected" represent the ratio and the number of loci detected by PacBio sequencing, respectively. **(C)** Methylation rates for HsdS_{A1-3} recognition motifs in WT, *pgdA*^D275N, *adr*^S438A and *adr*^rev strains.

sensed and relayed to the intracellular milieu to regulate *hsdS* inversions by an unknown signaling pathway. The first possibility is unlikely since our previous study showed a similar growth pattern between isogenic variants carrying *hsdS*_{A1} and other *hsdS*_A alleles [25]. Given the known role of NAM *O*-acetylation in resistance to the major autolysin LytA in *S. pneumoniae* [31], we tested the second possibility. We performed mutagenesis analysis of LytA, as well as other three PG hydrolases (LytB, LytC and CbpD), because these proteins also bind and cleave PG (Fig 4A) [9]. While deleting *lytB*, *lytC* or *cbpD* in the *adr*^S438A strain did not result in obvious impact on *hsdS* inversions, deleting *lytA* in *adr*^S438A led to the significant increase in *hsdS*_{A1}-carrying bacteria (Fig 4B). Reinstating the wildtype *lytA* sequence in the *adr*^S438A-Δ*lytA* mutant restored the low proportion of *hsdS*_{A1} configuration, similar to the parental strain *adr*^S438A. In a similar fashion, *adr*^S438A-Δ*lytA* mutant produced 100% O colonies (Fig 4C). In contrast, the *lytB*, *lytC* and *cbpD* mutants displayed a similar proportion of T/O ratio as the parental strain. These data strongly suggested that LytA functionally links *hsdS* inversions to colony phase, in response to the change of NAM *O*-acetylation state.

LytA is a cell wall hydrolase of *S. pneumoniae* that cleaves the amide bond between NAM and L-alanine in a zinc-dependent manner [32], which is responsible for autolysis at the stationary phase, during nutrient depletion or when PG synthesis is inhibited by antibiotics [33,34]. The amidase activity of LytA is also known to contribute to T colony

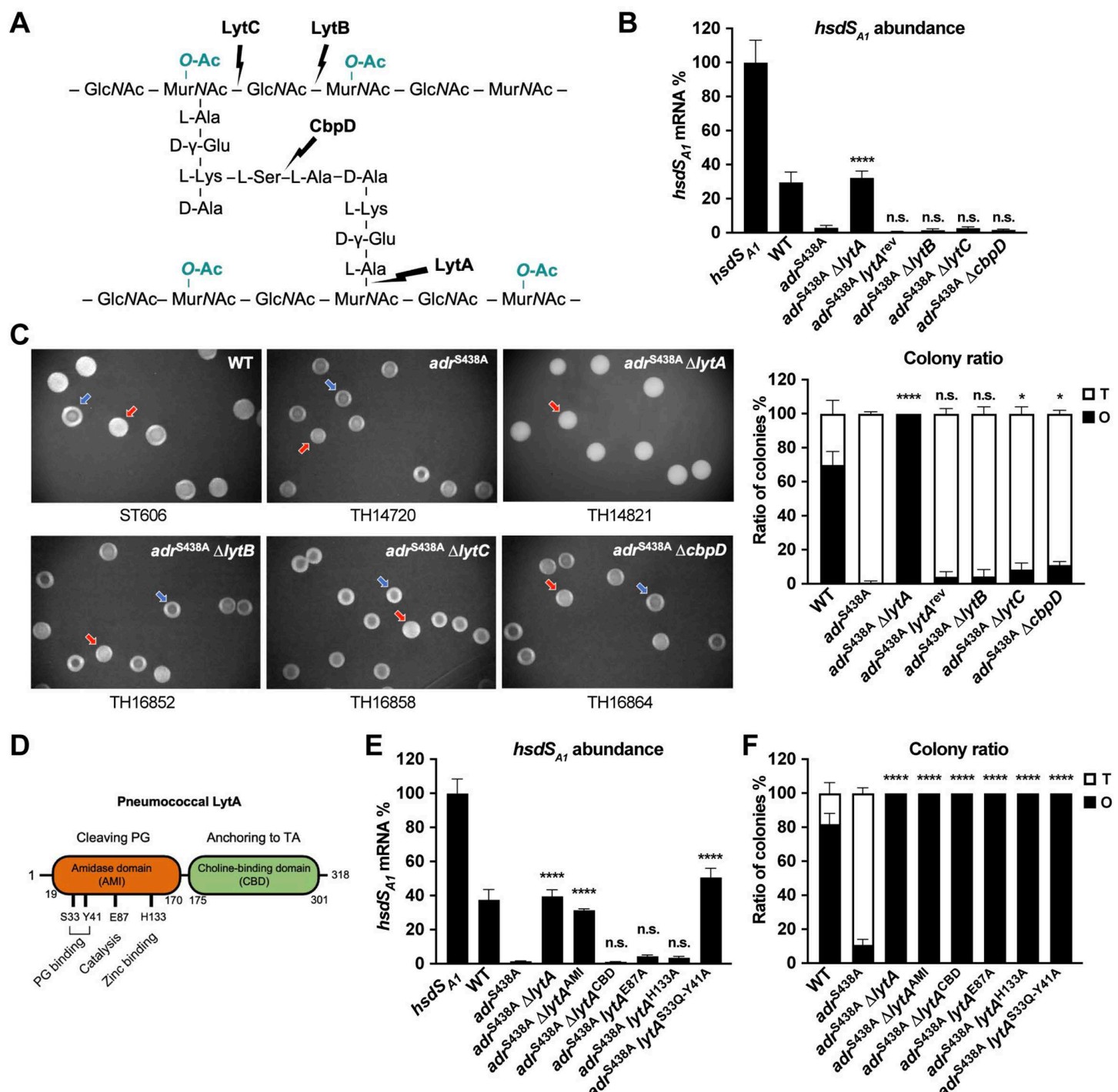

**Fig 4. The role of LytA in functionally linking decreased NAM *O*-acetylation with *hsdS* inversions and colony phase. (A)** Cell wall cleavage sites of pneumococcal PG hydrolases. MurNAc, *N*-acetylmuramic acid. GlcNAc, *N*-acetylglucosamine. **(B to C)** Relative abundance of the $hsdS_{A1}$ mRNA (B) and O/T colony phenotype (C) in *lytA*, *lytB*, *lytC* and *cbpD* mutants in the *adr*[S438A] background. Significance between *adr*[S438A] and other mutants is presented. **(D)** Illustration of the functional domains and amino acid residues of LytA. **(E to F)** Relative abundance of the $hsdS_{A1}$ mRNA (E) and O/T colony ratio (F) in different isogenic *adr-lytA* double mutants. Significance between *adr*[S438A] and other mutants is presented.

appearance [25]. To uncouple the *O*-acetylation-dependent and -independent impact of LytA on colony opacity, we constructed a *lytA* deletion mutant in WT strain. As reported previously [25], Δ*lytA* formed 100% O colonies (S3A and S3B Fig). However, Δ*lytA* had a similar level of $hsdS_{A1}$ mRNA with WT. This result indicates that, besides its direct impact on colony appearance by PG hydrolysis, LytA is able to modulate colony opacity in an *O*-acetylation-dependent manner.

LytA consists of an N-terminal catalytic amidase domain (AMI) and a C-terminal choline-binding domain (CBD) for protein anchoring to choline moiety of WTAs (Fig 4D). We assessed the specific contribution of the AMI and CBD domains to its function in modulating *hsdS* inversions in the *O*-acetylation-absent background. To our surprise, while the CBD-deficient strain ($adr^{S438A}$-Δ*lytA*$^{CBD}$) exhibited a similarly low proportion of $hsdS_{A1}$-carrying bacteria as the parental strain, the mutant without AMI ($adr^{S438A}$-Δ*lytA*$^{AMI}$) showed relatively higher representation of $hsdS_{A1}$ configuration (Fig 4E). By comparison, both the AMI and CBD mutants produced 100% O colonies, highlighting the dual functions of LytA (Fig 4F). This result suggested that the enzymatic domain of LytA is responsible for connecting the NAM *O*-acetylation status with the genetic configuration in the *cod* locus.

Several amino acid residues in the LytA amidase domain are responsible for the glycan-binding and catalytic, or zinc-binding activities (Fig 4D) [35,36]. We determined the roles of these activities in modulating *hsdS* inversions, in response to the loss of NAM *O*-acetylation, by selective change of representative residues. All the point mutants showed a comparable level of LytA as WT (S3C Fig). While mutating the two glycan-binding residues (33rd serine and 41st tyrosine) in $adr^{S438A}$ abolished the dominance of the $hsdS_{A1}$ configuration in the absence of NAM *O*-acetylation, the loss of the catalysis ($adr^{S438A}$-*lytA*$^{E87A}$) and zinc-binding ($adr^{S438A}$-*lytA*$^{H133A}$) residues did not yield obvious impact on the $hsdS_{A1}$ dominance in the same strain background (Fig 4E). On the other hand, all the three mutants produced 100% O colonies (Fig 4F). This result showed that the glycan-binding activity, but not enzymatic function, of LytA is essential for modulating *hsdS* inversions, while the PG hydrolysis by LytA is required for T colony formation.

## Multiple LytA-associated proteins contribute to NAM *O*-acetylation-dependent *hsdS* inversions

Considering that LytA functions in the extracellular milieu [34], it must engage its partner(s) to transmit the lack of NAM *O*-acetylation across the cell membrane to modulate *hsdS* inversions. Because the LytA glycan-binding activity is essential for modulating *hsdS* inversions in the absence of the NAM *O*-acetylation, we reasoned that WTA-anchored LytA is disengaged from PG when the C6-OH group of NAM is acetylated by the partner protein(s) due to the relatively higher affinity to LytA than PG; the lack of NAM *O*-acetylation makes PG more attractive to LytA than the partner protein(s); the LytA-less partner protein(s) activates the downstream signaling cascade to modulate *hsdS* inversions. We tested this hypothesis by comparing LytA-associated proteins via co-immunoprecipitation (Co-IP). Pneumococci expressing a Strep-tagged LytA were subjected to protein crosslinking with 1% formaldehyde before being lysed and incubated with biotin-conjugated beads to isolate proteins associated with LytA.

Mass spectrometry analysis identified 16 potential LytA-binding proteins under the NAM *O*-acetylation conditions (Fig 5A and S3 Table). Although these proteins were similarly expressed in two strains, they were more abundantly enriched in WT by Co-IP than in $adr^{S438A}$. Adr is also produced at the same level in two strains (S3D Fig). Some of these are intracellular proteins (e.g., ribosomal proteins and metabolic enzymes), which may interact with LytA before it is secreted or be contaminants. Based on the essential gene list of *S. pneumoniae* [37], we selectively characterized 7 genetically amendable hits. Deleting *ptvB* or *pcpA* in $adr^{S438A}$ yielded a similar phenotype as the Δ*lytA* mutant. The $adr^{S438A}$-Δ*ptvB* and $adr^{S438A}$-Δ*pcpA* strains showed significantly higher proportions of the $hsdS_{A1}$-positive bacteria (Fig 5B). By comparison, deleting the remaining 5 genes did not yield obvious effect.

PtvB was identified for its involvement in pneumococcal tolerance to vancomycin [38], but its precise function remains unclear. It contains a predicted short N-terminal transmembrane domain and a large extracellular segment (Fig 5C). In agreement with the requirement of PtvB for configuring *hsdS* inversion in the absence of NAM *O*-acetylation, PtvB is necessary for the T dominant phenotype of the $adr^{S438A}$ strain. The Δ*ptvB* mutant lost the T dominant genotype (Fig 5D) and

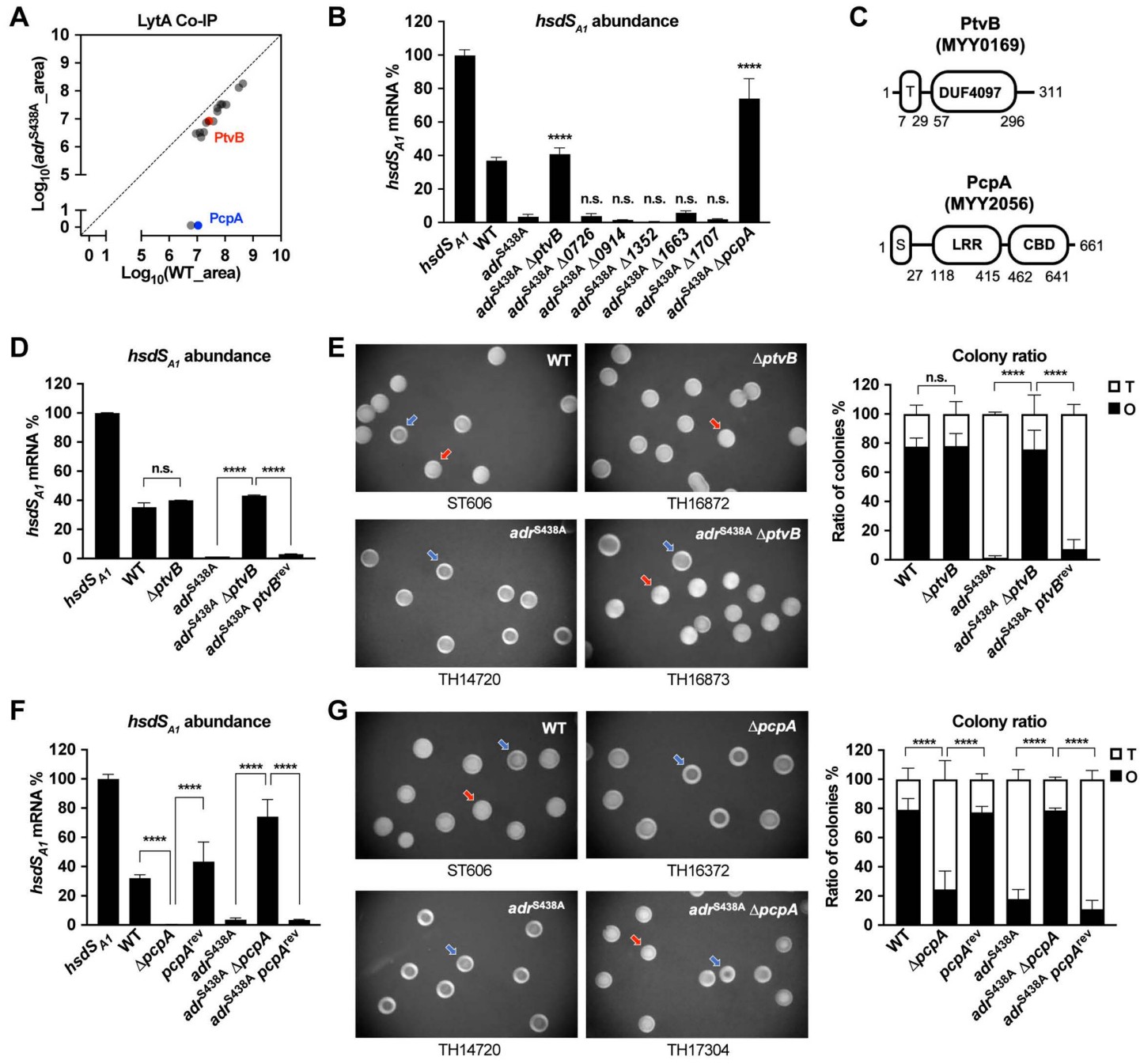

**Fig 5. The requirement of LytA-associated proteins for linking decreased NAM *O*-acetylation with the orientations of *hsdS* inversions. (A)** LytA-associated proteins enriched in the absence of NAM *O*-acetylation. The proteins pulled down from the lysates of ST606 and *adr*S438A strains by LytA-coated beads are presented as the average of the peak area obtained from four biological repeats in two individual experiments. **(B)** Relative abundance of the *hsdS*A1 mRNA in the mutants of LytA-associated proteins as in Fig 2C. Significance between *adr*S438A and other mutants is presented. **(C)** The domain structures of PtvB and PcpA. T, transmembrane region. S, signal peptide. LRR, leucine-rich repeat region. CBD, choline-binding domain. **(D to E)** Relative abundance of the *hsdS*A1 mRNA (D) and colony phenotypes (E) of *ptvB* mutants in WT and *adr*S438A backgrounds. **(F to G)** Relative abundance of the *hsdS*A1 mRNA (F) and colony phenotypes (G) of *pcpA* mutants in WT and *adr*S438A backgrounds.

phenotype (Fig 5E) of $adr^{S438A}$. This change was fully restored in revertant with the intact $ptvB$ ($adr^{S438A}$-$ptvB^{rev}$). In sharp contrast, the absence of PtvB did not affect $hsdS$ inversions and colony phenotype of WT. These experiments verified that PtvB promotes the non-$hsdS_{A1}$ orientation of $hsdS$ inversions only in the absence of NAM $O$-acetylation. However, we did not observe physical interaction between LytA and PtvB by bacterial adenylate cyclase-based two-hybrid (BATCH) experiment (S3E Fig), indicating that the functional linkage of the two proteins is bridged by an unknown partner. Furthermore, we tested potential collaboration of LytA and PtvB in promoting vancomycin tolerance based on the contribution of PtvB to pneumococcal tolerance to vancomycin [38]. Surprisingly, the loss of LytA itself made the wildtype bacteria more susceptible to vancomycin (S4 Fig), which made it difficult to interpret the functional connection between LytA and PtvB in vancomycin tolerance.

While PcpA is reported as a virulence factor [39,40], and a potential vaccine candidate [41,42], its physiological function remains to be defined. It contains a C-terminal leucine-rich repeat region and a N-terminal choline-binding domain (Fig 5C) [43]. Our additional experiment also confirmed that PcpA is involved in regulating $hsdS$ inversions (Fig 5F). In the $adr^{S438A}$ background, deleting $pcpA$ led to the return of the T-dominant phenotype to the O-dominant level of WT, which was completely restored in $pcpA$ revertant (Fig 5G). Surprisingly, unlike the deletion of $ptvB$, the absence of $pcpA$ in WT background significantly affected $hsdS$ inversions and colony opacity as well. The proportions of O colonies and $hsdS_{A1}$-carrying bacteria were significantly reduced in the $\Delta pcpA$ mutant (Fig 5F and 5G). This indicated that PcpA plays a more complex role than PtvB in regulating $hsdS$ inversions.

## PtvB modulates $hsdS$ inversions by interacting with PtvC and DimA

Because PtvB is predicted to be primarily localized in the extracellular space with a transmembrane segment and a short intracellular tail of six amino acids, PtvB is unlikely to directly modulate $hsdS$ inversions. We identified pneumococcal protein(s) that interacts with PtvB in the absence of NAM $O$-acetylation. The PtvB-associated proteins were enriched by incubating biotin-conjugated beads with the lysates of the Strep-PtvB expressing WT and $adr^{S438A}$ strains. Mass spectrometry revealed 39 proteins that were selectively enriched in $adr^{S438A}$ as compared with WT (Fig 6A and S4 Table). Since many of these proteins might be contaminants being accidentally crosslinked to PtvB or other proteins, we verified the functional involvement of 16 genetically amendable genes in $hsdS$ inversion regulation by mutagenesis in $adr^{S438A}$. Only $\Delta hsdM$ and $\Delta myy1025$ showed significant impact on $hsdS$ inversion in $adr^{S438A}$ as $\Delta ptvB$, in terms of $hsdS_{A1}$-carrying bacteria, whereas deleting the other 14 genes led to marginal or no effect (Fig 6B). The $hsdM$ encodes the DNA methyltransferase subunit of the type I R-M system in the $cod$ locus, and is located immediately upstream of $hsdS_A$ [25]. Since the enrichment of HsdM could be explained by potential transcriptional upregulation of the $hsdRMS_A$ operon [26], we chose to focus on $myy1025$.

The $myy1025$ encodes a cytoplasmic protein of 424 amino acids without any characterized or predicted function. We renamed it as DNA inversion modulator A, $dimA$. Further experiments showed that the impact of DimA on the $hsdS$ configuration was reversed with the intact $dimA$ in $adr^{S438A}$ (Fig 6C). In addition, deleting $dimA$ in WT did not yield significant change in the $hsdS$ configuration. Consistent with its impact on $hsdS$ inversions, deletion of $dimA$ in $adr^{S438A}$ ablated the O colony-dominant phenotype, but removing $dimA$ in WT did not yield obvious impact on the colony phenotype (Fig 6D). These experiments showed that the PtvB-associated DimA exerts an essential role in linking NAM $O$-acetylation and DNA inversions in the $cod$ locus.

We next characterized the physical interaction between PtvB and DimA using BATCH assay. The T18 fragment of adenylate cyclase was fused to DimA at either N- and C-terminus, and co-expressed with T25-tagged PtvB at its C terminus. While positive control displayed expected colony color and β-galactosidase activity, the DimA fusions did not display significant change in the presence of T25-PtvB (Fig 6E). This result suggested the lack of direct interaction between PtvB and DimA, which may be attributed to the extremely short cytoplasmic tail of PtvB.

PtvB is predicted to form a transmembrane complex with PtvA and PtvC, both of which are encoded by the same $ptv$-$ABC$ operon (S5A Fig) [38]. Accordingly, the colony and β-galactosidase results demonstrated positive interactions of PtvB

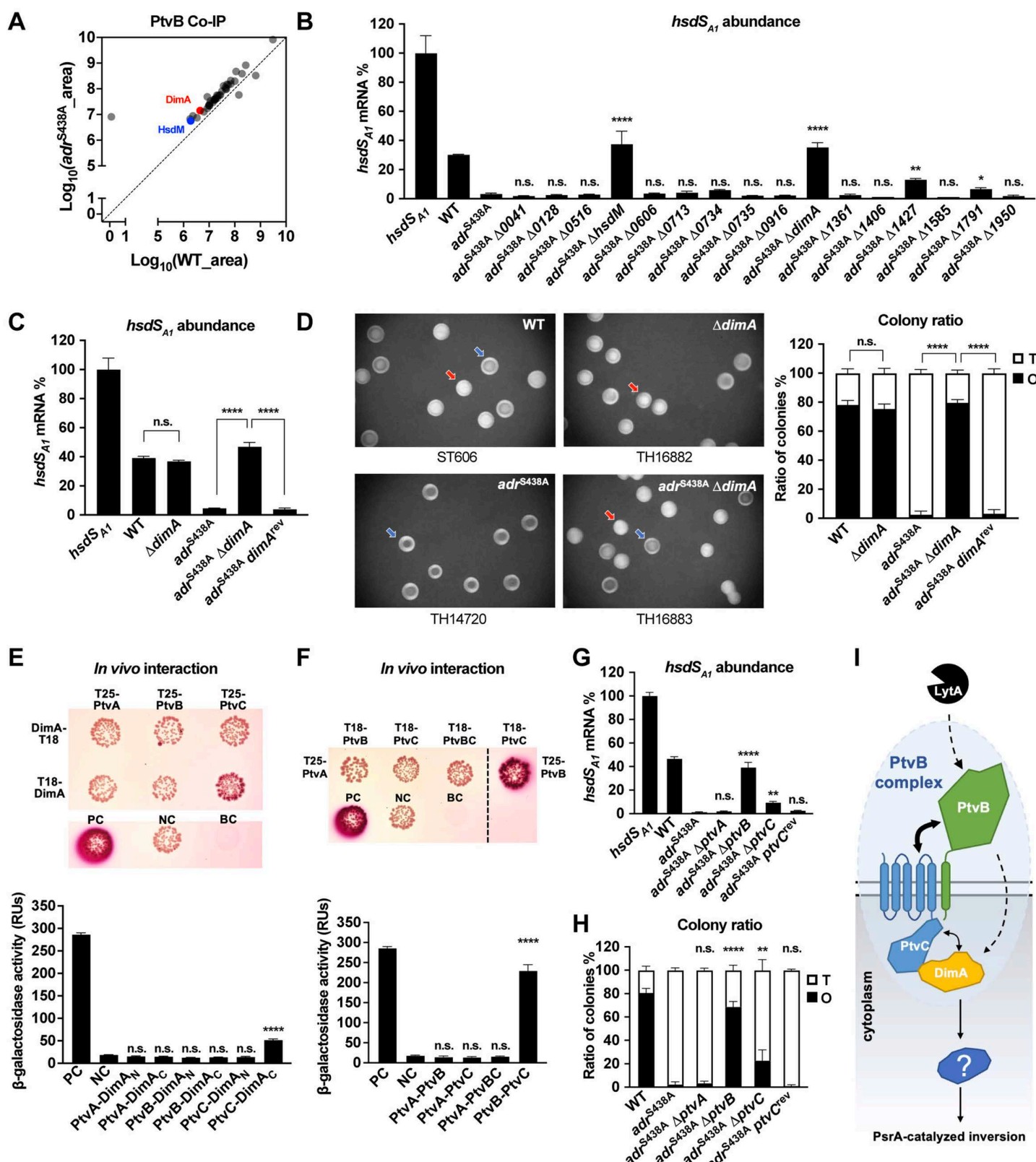

**Fig 6. Functional relationship of DimA and PtvC with PtvB. (A)** PtvB-associated proteins enriched by PtvB-coated beads from the lysates of ST606 and $adr^{S438A}$ strains are presented as in Fig 5A. **(B)** Relative abundances of the $hsdS_{A1}$ mRNA in the mutants of PtvB-associated proteins are shown as

in Fig 2C. Significance between $adr^{S438A}$ and other mutants is presented. **(C to D)** Relative abundance of the $hsdS_{A1}$ mRNA **(C)** and colony phenotypes **(D)** of $dimA$ mutants. **(E)** Detection of DimA interactions with PtvA, PtvB and PtvC by bacterial two-hybrid assay. Colonies on the MacConkey/maltose plates (upper panel) and β-galactosidase activity (lower panel) are shown for each reporter strain. PC, positive control (pKT25-*zip* and pUT18C-*zip*); NC, negative control (empty vectors pKT25 and pUT18C); BC, blank control without plasmid. Significance between NC and experimental groups is presented. **(F)** Detection of physical interactions among PtvA, PtvB and PtvC by bacterial two-hybrid assay. The data are shown as in **E**. **(G to H)** Relative abundance of the $hsdS_{A1}$ mRNA **(G)** and colony phenotypes **(H)** of $ptvA$, $ptvB$ or $ptvC$ mutants in $adr^{S438A}$ background. Significance between $adr^{S438A}$ and other mutants is presented. **(I)** The protein interaction model of LytA and PtvB to modulate $hsdS$ inversions. The thickness of the double-headed arrows represents the interaction intensity.

with PtvC but not with PtvA (Fig 6F). Since PtvC contains a relatively longer cytoplasmic domain (173 amino acids) than PtvB, we tested the likelihood that PtvC interacts with DimA. DimA with the N-terminal T18 showed positive colony color and significant level of β-galactosidase when being co-expressed with T25-tagged PtvC (Fig 6E). However, the C-terminal tagged DimA did not show obvious interaction with PtvC (S5B Fig). This result indicated that cytoplasmic protein DimA indirectly interacts with membrane-bound PtvB through its partner PtvC. Lastly, we verified the functional role of PtvC in regulating $hsdS$ inversions by deleting $ptvC$ in $adr^{S438A}$. The $adr^{S438A}$-$\Delta ptvC$ mutant showed significant increase in $hsdS_{A1}$-carrying bacteria (Fig 6G) and O colonies (Fig 6H). These phenotype of $adr^{S438A}$-$\Delta ptvC$ were restored to the parental level in $ptvC$ revertant. It is obvious that the functional impact of $ptvC$ deletion on $hsdS$ inversions was much less pronounced than $ptvB$ or $dimA$ deletion. This result suggested that PtvC partially contributes to the PtvB-mediated regulation of $hsdS$ inversions (Fig 6I).

### The lack of NAM *O*-acetylation triggers the up-regulation of the invertase PsrA

To define how DimA modulates the PsrA-catalyzed inversions, we assessed its potential interaction with PsrA by BATCH assay. As shown in Fig 7A, the experiment did not show any significant direct interaction between these two proteins. This result indicated that DimA regulates $hsdS$ inversions without direct interaction with PsrA.

To further identify the functional connection between NAM *O*-acetylation and $hsdS$ inversions, we determined pneumococcal transcriptomes in the presence (WT strain) or absence ($adr^{S438A}$) of NAM *O*-acetylation by RNA sequencing (RNA-seq). Pairwise comparison revealed 215 genes with at least 1.5-fold transcriptional change by the loss of the Adr acetyltransferase activity (Fig 7B and S5 Table). Likewise, RNA-seq comparison between $adr^{S438A}$ and $adr^{S438A}$-$lytA^{S33Q-Y41A}$ also identified 249 genes with at least 1.5-fold changes in transcription in the absence of the glycan-binding activity of LytA (Fig 7B and S6 Table). To simplify these complex data, we reasoned that the factor(s) linking NAM *O*-acetylation and $hsdS$ inversions should be commonly affected by the loss of the acetyltransferase of Adr (WT vs. $adr^{S438A}$) and glycan-binding activity ($adr^{S438A}$ vs. $adr^{S438A}$-$lytA^{S33Q-Y41A}$). Along this line, we identified nine genes whose transcription was significantly altered under the two conditions (Fig 7C). $psrA$ and $hsdS_C$ were the two genes with significant upregulation both in the lack of NAM *O*-acetylation and the glycan-binding activity of LytA. The $psrA$ mRNA was increased by 1.7-fold in $adr^{S438A}$ as compared with WT. To a greater extent, there were 5.5-fold more $psrA$ transcripts in $adr^{S438A}$ than that in $adr^{S438A}$-$lytA^{S33Q-Y41A}$. The expression of $hsdS_C$, which is located immediately upstream of $psrA$ in the $cod$ locus (Fig 2A), was up-regulated by at least 3-fold in $adr^{S438A}$. Additionally, we observed modest transcriptional down-regulation in seven genes, including $hsdS_B$. $hsdS_B$ is located at the immediate downstream of $psrA$ (Fig 2A), but is transcriptionally separated from $psrA$ by a transcriptional terminator [25].

In the context of our previous observation that overexpression of $psrA$ leads to enrichment of the transparent non-$hsdS_{A1}$ allelic configurations [29,44]. We next focused on verifying the expression of $psrA$ under various strains by quantitative RT-PCR. As compared with WT, $adr^{S438A}$ showed a modest but significant increase in $psrA$ transcription (Fig 7D). In a consistent manner, $psrA$ expression was reduced to the WT level in the absence of LytA glycan-binding activity. These data suggested that the LytA-mediated regulatory circuit modulates $hsdS$ inversions at least in part by transcriptional upregulation of $psrA$. We also observed that the enhanced $psrA$ transcription in $adr^{S438A}$ was reduced to the WT level by

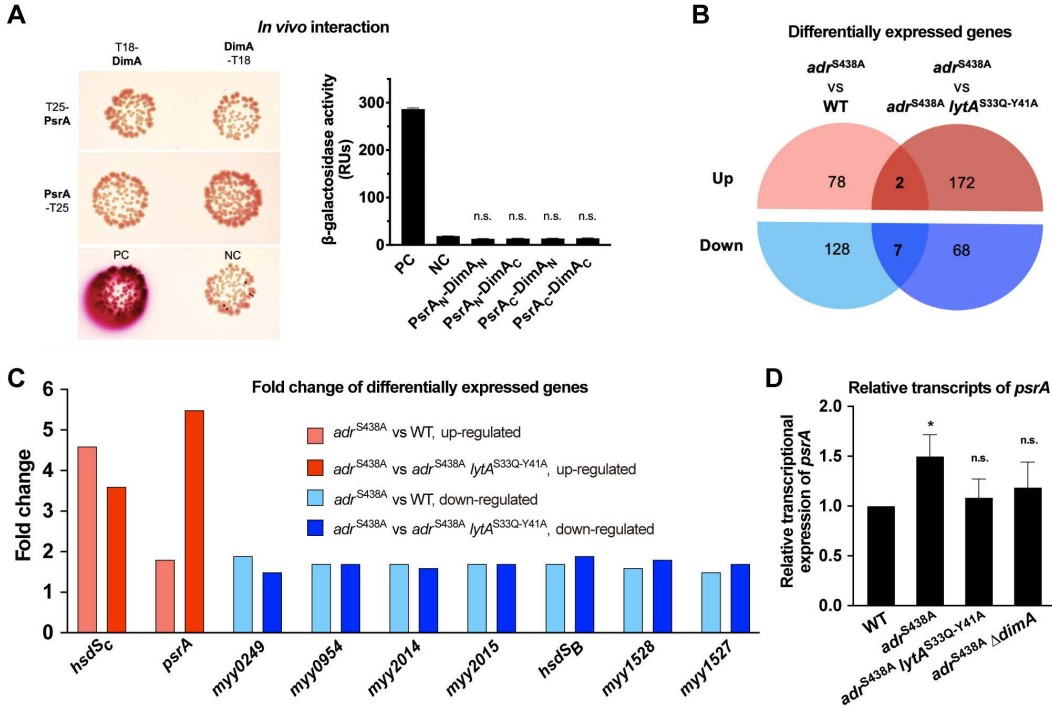

**Fig 7. Upregulation of *psrA* transcription in the absence of NAM *O*-acetylation. (A)** Detection of interactions between DimA and PsrA by bacterial two-hybrid assay. Colonies on the MacConkey/maltose plates (left panel) and β-galactosidase activity (right panel) are shown for each reporter strain as in Fig 6E. Significance between NC and experimental groups is presented. **(B)** Venn diagram of differentially expressed genes in WT and *adr-lytA* mutant relatively to *adr*^S438A. Genes were sorted by a cut-off value of fold change ≥1.5 and $P_{adj}$<0.05. **(C)** Fold change of nine differentially expressed genes in the mRNA level of WT and *adr-lytA* mutant relatively to *adr*^S438A. **(D)** The transcription of *psrA* in the *adr*^S438A mutants was detected by qRT-PCR.

removing *dimA* (Fig 7D), suggesting DimA somehow impacts the transcription of *psrA* in the absence of NAM *O*-acetylation. However, the linkage between DimA and the regulation of *psrA* remains to be defined. Together, these results suggested that, in response to the absence of NAM *O*-acetylation, LytA modulates epigenetic and cellular phases of *S. pneumoniae* at least in part by transcriptional upregulation of *psrA* through a multi-component signaling circuit.

## Acetyl-CoA may be a linker between nutrient availability and NAM *O*-acetylation

Jones *et al.* have shown that NAM *O*-acetyltransferase A (OatA) of *Staphylococcus aureus* uses acetyl-CoA as the donor of the acetyl group to modify NAM [45]. Based on the high sequence identity between OatA and Adr, acetyl-CoA likely acts as the substrate of pneumococcal Adr. Acetyl-CoA of *S. pneumoniae* is primarily produced by pyruvate formate lyase encoded by *pfl* and pyruvate dehydrogenase complex (PDHC) (Fig 8A) [46,47]. PDHC is encoded by *acoA*, *acoB*, *acoC* and *acoL* (S6A Fig). We tested potential impact of acetyl-CoA availability on NAM *O*-acetylation in the absence of pyruvate formate lyase (Δ*pfl*) or PDHC (Δ*acoB*). Our repeated attempts to construct a double mutant were unsuccessful, likely due to synthetic lethality. Both the Δ*acoB* and Δ*pfl* strains showed significant reduction in acetyl-CoA (Fig 8B). In a consistent pattern, the level of NAM *O*-acetylation was also significantly reduced in Δ*acoB* and Δ*pfl*, as compared with that in WT (35.7%) (Fig 8C). This result showed that cellular acetyl-CoA level greatly impacts NAM *O*-acetylation.

To test whether the cellular acetyl-CoA level can indirectly modulate *hsdS* inversions, we measured the ratio of *hsdS*~A1~-carrying bacteria under the acetyl-CoA deficient conditions. The Δ*acoB* mutant showed significant reduction

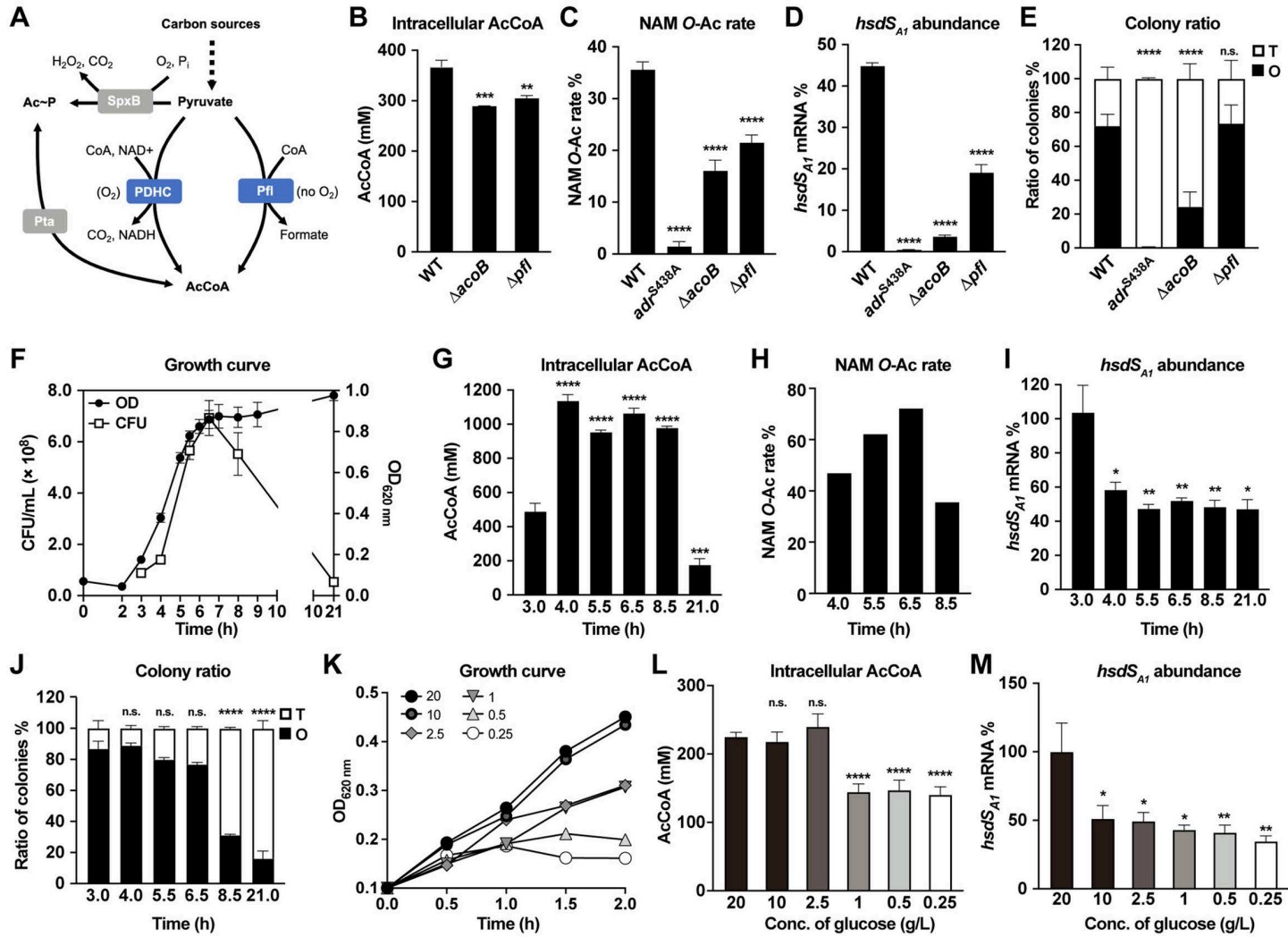

**Fig 8. The impact of acetyl-CoA on NAM *O*-acetylation and colony phase. (A)** The synthesis of acetyl-CoA (AcCoA) and acetyl phosphate (Ac~P) from carbon sources. Pyruvate oxidase SpxB is responsible for the synthesis of Ac~P. The pyruvate dehydrogenase complex (PDHC) and pyruvate formate lyase (Pfl) are the functional enzymes to produce AcCoA under aerobic and anaerobic conditions, respectively. **(B to E)** The intracellular acetyl-CoA amount **(B)**, NAM *O*-acetylation level **(C)**, relative abundance of the *hsdS*$_{A1}$ mRNA **(D)**, and colony ratios (E) of Δ*acoB* (TH17288) and Δ*pfl* (TH17290) are shown as mean ± s.d. of 3-4 replicates in a representative experiment. **(F)** Growth curve of ST606 in TSB medium. **(G to J)** The intracellular acetyl-CoA amount **(G)**, NAM *O*-acetylation level **(H)**, the relative abundance of the *hsdS*$_{A1}$ mRNA **(I)**, and colony ratio **(J)** of ST606 pneumococci at various time points cultured in TSB medium. The values at 3 h were used as references for statistical comparison. **(K)** Growth curve of ST606 in CDM with different concentrations of glucose. **(L to M)** The intracellular acetyl-CoA amount **(L)** and relative abundance of the *hsdS*$_{A1}$ mRNA **(M)** of ST606 bacteria collected at 1.5 h in CDM with different concentrations of glucose. The values at 20 g/L glucose were used as references for statistical comparison.

in the ratio of *hsdS*$_{A1}$-carrying bacteria as compared with WT (Fig 8D). To a lesser extent, the *hsdS*$_{A1}$-positive bacteria were also significantly decreased in Δ*pfl*. Phenotypically, Δ*acoB* exhibited significant decrease in the proportion of O colonies, whereas the Δ*pfl* did not show obvious phenotype (Fig 8E and S6A Fig). The more severe impact of *acoB* deletion on *hsdS* inversions agrees with the dominant role of PDHC in pneumococcal acetyl-CoA biosynthesis under aerobic conditions [46]. These results revealed that cellular acetyl-CoA availability substantially influence the extent of NAM *O*-acetylation, and indirectly modulates *hsdS* inversions.

As illustrated in Fig 8A, pyruvate is the main precursor of acetyl-CoA in *S. pneumoniae* [46,48]. It is thus reasonable to predict that the nutrient availability determines the level of cellular acetyl-CoA. To test this possibility, we cultured WT bacteria to the lag, exponential and stationary phases, and measured viable bacterial counts (colony forming unit, CFU/ml) (Fig 8F), and corresponding acetyl-CoA levels (Fig 8G) at various time points post inoculation. As compared with bacteria at the lag phase (hr 3), the cells at the logarithmic phase (4, 5.5 and 6.5h) showed a maximal level of acetyl-CoA (Fig 8G). Cellular acetyl-CoA remained at a high level at the early death phase (8.5h), and dropped at the late death phase (21h). NAM *O*-acetylation was also found to change in a growth phase-dependent manner (Fig 8H). The growth phase-dependent dynamics of cellular acetyl-CoA and NAM *O*-acetylation indicates the correlation among nutrient availability, the level of cellular acetyl-CoA, and cell wall *O*-acetylation.

To determine the impact of growth phase on *hsdS* inversions, we quantified the relative proportion of *hsdS*$_{A1}$-carrying bacteria at different growth phases. Virtually bacteria possessed the *hsdS*$_{A1}$ allele in the *cod* locus at the early logarithmic phase (3h) (Fig 8I). However, the proportional abundance was steadily decreased from 58.3% at 4h to 47.0% at 21h. At the phenotypic level, the proportion of O colonies gradually decrease from the mid logarithmic phase to the late dying period (Fig 8J). These data suggested that the extent of NAM *O*-acetylation reflects environmental conditions, including nutrient availability, reduced pH and accumulation of toxic metabolites.

We finally assessed the impact of carbon source on acetyl-CoA by growing pneumococci in a chemical defined medium (CDM) with various concentrations of glucose. Consistent with the glucose concentration-dependent growth (Fig 8K), there was a concentration-dependent reduction both in the level of acetyl-CoA abundance (Fig 8L) and the proportion of *hsdS*$_{A1}$-carrying bacteria (Fig 8M) at 1.5h post inoculation, at which the impact of glucose concentration on bacterial growth became obvious. This change was not due to glucose-dependent change of *adr* transcription because the *adr* mRNA level did not show significant change under various glucose concentrations (S6B Fig). These data support the notion that cellular nutrient availability impacts the *hsdS* configurations.

## Discussion

NAM *O*-acetylation is important for pneumococcal resistance to lysozyme- and LytA-catalyzed cell wall hydrolysis. This work has revealed that the extent of NAM *O*-acetylation defines the *hsdS* gene configurations in the *cod* locus, genome methylation patterns and colony phases. As illustrated in Fig 9, the existing data prompt us to propose a working model to explain this regulatory process. When the C6-OH groups of NAMs in the cell wall are heavily acetylated, the *cod* locus adopts the *hsdS*$_{A1}$ allelic configuration, which leads to the methylation of nearly all 2,060 sites of the HsdS$_{A1}$ motif in pneumococcal genome, and the formation of an O colony-dominant population. In the absence of NAM *O*-acetylation, LytA binds to PG and thereby activates the downstream regulatory steps to indirectly modulate the orientations of *hsdS* inversion towards the *hsdS*$_{A3}$-dominant configurations. The lack of methylation at the HsdS$_{A1}$ motif sites leads to the formation of a T colony-dominant population. At the physiological level, the extent of NAM *O*-acetylation appears to reflect the nutrient-dependent status of cellular acetyl-CoA, the donor of the acetyl group for NAM *O*-acetylation. In short, our data support the postulation that *S. pneumoniae* uses NAM *O*-acetylation as an extracellular signal of cellular metabolism/nutrient supply to synchronize bacterial metabolism and growth according to nutrient availability in host niches. Given the fact that NAM *O*-acetylation and the *hsdS* inversion systems are prevalent in many bacteria [13,49], the functional linkage between NAM *O*-acetylation and the epigenetic machinery may operate in other bacteria.

With the guidance of our serendipitous observation in our earlier study [28], we found a causal relationship between low NAM *O*-acetylation and colony phase in *S. pneumoniae*. Based on the fact that the orientation of *hsdS* inversions in the *cod* locus [25,27], we further showed that absence of NAM *O*-acetylation determines pneumococcal methylome by modulating *hsdS* inversions. The acetylase-negative *adr* mutant lost the ability to produce the O colony-dominant populations; instead, the mutant uniformly produced T colonies. Additional experiments revealed the complete absence of the HsdS$_{A1}$-specified genome methylation in the *adr* mutant, which is consistent with the loss of the O colony-defining *hsdS*$_{A1}$ allele in the *cod* locus. While the cross-membrane molecular communications have been well documented in bacteria, to

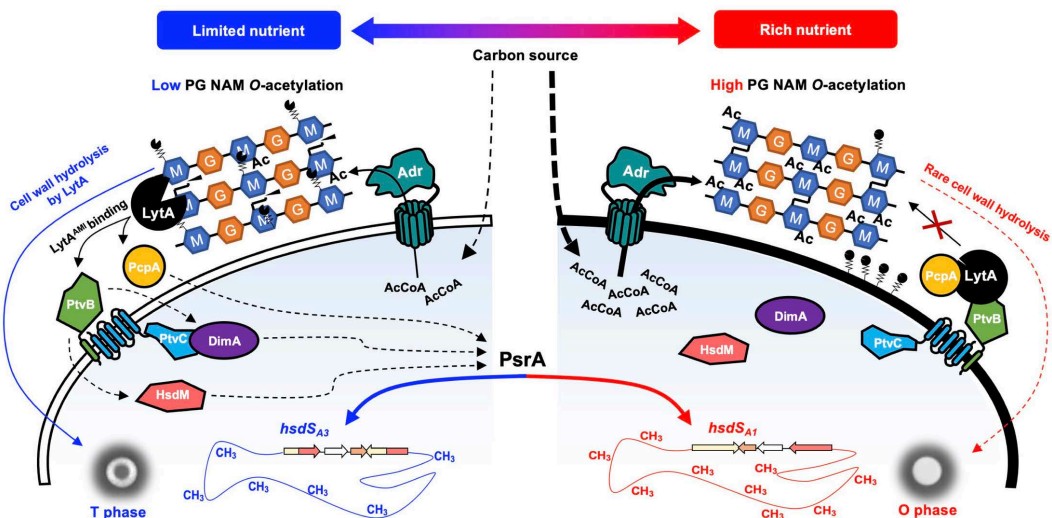

**Fig 9. A working model for the regulatory function of NAM *O*-acetylation in pneumococcal adaptation to nutrient availability.** Variation of carbon sources in different host niches impacts intracellular acetyl-CoA (AcCoA) concentration and thereby NAM *O*-acetylation level. The absence of NAM *O*-acetylation increases the affinity of PG to LytA, which activates the cross-membrane signal pathway that requires PcpA, PtvB, PtvC, HsdM and DimA. By an uncharacterized mechanism, DimA modulates PsrA-catalyzed *hsdS* inversions, genome methylation pattern and colony phase. Ac, acetyl. M, NAM. G, NAG. Polygonal lines attached to NAM and cell membrane represent WTA and LTA, respectively.

the best of our knowledge, this is the first report of a functional connection between the states of bacterial cell wall structure and genome methylation.

Our data strongly suggest that LytA is an extracellular sensor of NAM *O*-acetylation for intracellular epigenetic responses. Based on the functional connection between NAM *O*-acetylation and LytA-mediated PG hydrolysis [17], our mutagenesis analysis of the four known cell wall hydrolases led to the identification of LytA as the cell wall-associated molecular sensor for intracellular epigenetic responses. In agreement with the previous findings that NAM *O*-acetylation interferes with the binding and enzymatic actions of LytA toward pneumococcal cell wall [17], we found that the modification also blocks LytA from modulating *hsdS* inversions. This conclusion is supported by our observation that mutating the LytA PG-binding residues in the *adr*[S438A] mutant abolished the impact of the lack of NAM *O*-acetylation on *hsdS* inversions, but deletion of CBD or point mutations of catalysis-associated residues did not affect the *hsdS* gene configuration of the *adr*[S438A] mutant. While it remains to be determined how LytA interacts with peptidoglycan, WTA and its protein partners in the extracellular milieu, this work has uncoupled the LytA activity in modulating *hsdS* inversions from its enzymatic activity in cell wall hydrolysis (for autolysis and T colony formation). In this context, it is important to consider how *S. pneumoniae* coordinates covalent anchoring of WTAs to the C6-OH group of NAM with the Adr-catalyzed *O*-acetylation at the same substrate. Flores-Kim *et al.* have demonstrated that LytA-catalyzed autolysis is activated by dominant place teichoic acids to the cell wall during stationary phase or after penicillin treatment [12]. It is possible that shrinking supply of acetyl-CoA due to nutrient depletion reduced the extent of NAM *O*-acetylation, and thereby left more free C6-OH groups for anchoring WTA and LytA to PG. However, the LytA-mediated modulation of *hsdS* inversions occurs in the early exponential phase when LytA-driven autolysis does not occur. Moreover, the glycan-binding activity of LytA is essential for modulating *hsdS* inversions, but the other essential activities for autolysis (catalysis and choline-binding) are not necessary (see Fig 4). These lines of information argue that the regulatory function of LytA operates independent of its cell hydrolysis activity.

This work reveals that multiple proteins are required for LytA to transduce the signal of the lack of NAM *O*-acetylation across the cell membrane to modulate *hsdS* inversions. PtvB and PcpA, two pneumococcal proteins without any defined functions, were found to modulate *hsdS* inversions along with LytA. PcpA is non-covalently anchored to the choline

residues of cell wall teichoic acids via its C-terminal choline-binding domain, and thereby physically linked with the cell wall [43,50]. The N-terminal leucine-rich region of PcpA may directly or indirectly interact with LytA, because the leucine-rich repeat structures have been shown to be involved in protein-protein interactions in many other organisms [51,52]. However, the essential role of PcpA in modulating *hsdS* inversions seems to be unrelated to NAM *O*-acetylation, since the *pcpA* mutant abolished the impact of low NAM *O*-acetylation on *hsdS* inversions, but also reduced the ratio of $hsdS_{A1}$-carrying bacteria with normal NAM *O*-acetylation. The precise mechanisms of cell wall-associated PcpA actions await further investigation.

PtvB is encoded by the highly conserved vancomycin-inducible *ptvABC* operon, and predicted to form a membrane-associated protein complex with PtvA and PtvC [38]. Consistently, our bacterial two-hybrid assay revealed a strong physical interaction between PtvB and PtvC. Our further Co-IP and bacterial two-hybrid experiments showed physical interaction of PtvC with cytoplasmic protein DimA, a protein without any known function. However, the phenotype of *ptvC* mutant was subtle compared to *ptvB* mutant, suggesting there is an alternative way that connects PtvB and DimA. These data have thus uncovered a protein trail for the LytA-mediated functional linkage between absence of NAM *O*-acetylation and *hsdS* inversions, in which the extracellular cell wall-associated (LytA and PcpA), membrane-bound (PtvB and PtvC) and cytoplasmic (DimA) proteins are all required. While it remains to be determined how the proteins are functionally orchestrated to achieve the signaling function, the existing lines of evidence support a working model that will guide future investigations (Fig 9). Specifically, under the NAM *O*-acetylation condition, LytA is fended off from the *O*-acetylated PG substrate, and complexed with cell wall-associated protein(s) (e.g., PcpA and/or PtvB), which prevents both the cell wall hydrolysis (required for enzymatic formation of T colonies) and the cross-membrane signaling (required for altering the orientations of *hsdS* inversions toward the non-$hsdS_{A1}$ allelic configurations), leading to the formation of O colonies phenotypically. In an opposite manner, when the C6-OH groups of NAM residues are non-acetylated, they become more attractive to cell wall-associated LytA than the partner protein(s); detachment of LytA from the molecular complex initiates a cross-membrane signaling cascade that modulates PsrA-catalyzed *hsdS* inversions and pneumococcal methylome by an unknown mechanism(s).

DNA inversions are highly prevalent in both prokaryotic and eukaryotic organisms [53]. A recent study shows that the orientations of reversible genomic DNA sequences in gut microbiota are modulated by bacteriophages and host inflammation [54]. However, the molecular mechanisms governing the orientations of the inversion reactions are only extensively studied in bacteriophage λ integrases, which require accessory host factors including IHF, Fis, and Xis [55]. For instance, in the Hin recombinase-catalyzed inversions of flagellin genes in *Salmonella* Typhimurium, the orientations of the two invertible genome sequences encoding two flagellin proteins are regulated by the DNA-binding proteins Fis (factor for inversion stimulation) and/or HU [53]. In particular, HU loops the invertible sequence, whereas Fis enhances the assembly of the supercoiling-dependent invertasome by binding to an enhancer sequence within the invertible sequence [5,56]. While PtvB, PtvC and DimA are necessary for this signaling line, it is unlikely that these proteins directly interact with the invertible sequences in the *cod* locus, because PtvB, PtvC and DimA don't contain any detectable DNA-binding sequences. Our transcriptional analysis suggests that the LytA-mediated signaling circuit modulates the orientations of *hsdS* inversions by upregulation of the invertase PsrA. The detail of the regulatory process requires further investigations.

The LytA signaling pathway and *hsdS*-targeting TCSs may share certain common metabolic features. One of the common features is the dominate impact on pneumococcal methylome. The $adr^{S438A}$ mutant completely lost the 6-mA methylation of the $HsdS_{A1}$-specific DNA motif in pneumococcal genome, but displayed significantly increased methylation of $HsdS_{A3}$-specific motifs. This pattern of the $HsdS_{A1}$-OFF genome methylation and the $hsdS_{A1}$-OFF orientation in the *cod* locus resembles what was previously observed with the mutants of the four TCSs (TCS06, TCS08, TCS09 and TCS11) [28]. Especially, the *O*-acetylation-deficient strain shares a striking similarity with the mutant of TCS06 in methylome and *hsdS* configuration. PacBio sequencing revealed no 6-mA methylation for any of the 2,060 $HsdS_{A1}$ recognition sites in the deletion mutant of the *rr06* gene encoding the response regulator of TCS06 although virtually all the sites were methylated

in parental strain. Likewise, the *rr06* mutant also displayed 6-mA methylation for nearly all of the 1,472 HsdS$_{A3}$ recognition sites [28]. TCS06 activates the transcription of *cbpA* encoding choline-binding protein A (CbpA), a cell wall-associated protein with multiple functions in pneumococcal pathogenesis, but the environmental signal(s) sensed by the system remains undefined [57–59]. These lines of evidence have uncovered that multiple extracellular signals modulate *hsdS* inversions to stabilize the opaque-ON *hsdS$_{A1}$* orientation and thereby the HsdS$_{A1}$-driven methylome.

The functional convergence of the *hsdS*-targeting LytA signaling pathway and TCSs has multiple implications in *hsdS* inversion regulation and biological functions of HsdS$_{A1}$-driven methylome. While it is currently unknown how the orientation of *hsdS* inversion reactions is controlled, these signaling systems may utilize a common downstream mechanism(s) to modulate the orientation of *hsdS* inversion reactions, although the upstream signal processes must be unique for each signaling system. The LytA-associated proteins identified in this work will be instrumental for defining the system-specific and common details of *hsdS* inversion regulation. Moreover, the LytA signaling pathway and the *hsdS*-targeting TCSs may drive certain common epigenetic/cellular responses because they all target the same *hsdS* inversion locus. This notion is supported by the similar T colony-dominant phenotype among the mutants lacking in NAM *O*-acetylation or the *hsdS*-targeting TCSs, respectively. While colony opacity is the best characterized phenotype that is defined by the HsdS$_A$-driven methylome, the *hsdS* configuration and resulting methylomes should have profound impact on pneumococcal biology. Therefore, the LytA signaling pathway is a potential breakthrough in understanding the functions of the pneumococcal epigenetic regulatory machinery.

The acetylation status at the C6-OH groups of NAM residues may be an extracellular indicator of nutrient/metabolic condition. In agreement with the essential role of acetyl-CoA in post-synthetic NAM *O*-acetylation [30,60], our data showed that the acetyl-CoA level dynamics is associated with the level of NAM *O*-acetylation and the orientation of *hsdS* inversions. In the context of the causal relationship between NAM *O*-acetylation and *hsdS* inversions, this study argues that acetyl-CoA indirectly modulates *hsdS* inversions via the extracellular loop via NAM *O*-acetylation, although our data cannot exclude the possibility that acetyl-CoA also modulates *hsdS* inversions through a NAM *O*-acetylation-independent mechanism(s). Along this line, the state of NAM *O*-acetylation may represent an extracellular signal of cellular acetyl-CoA status (or glycan supply) for pneumococcal adaptation to various host niches. In particular, glucose and many other nutrients are rich in the blood (during invasive infection), but are much less available at the nasopharynx, the natural colonization niche of *S. pneumoniae* [61]. This difference in glucose is consistent with our previous observation that *S. pneumoniae* mostly synthesizes methionine for the survival in the upper airway of mice, but switch to take up the amino acid during blood infection [62]. Under the poor nutrient conditions (e.g., the upper airway of healthy humans), the shortage of acetyl-CoA leads to relatively lower levels of NAM *O*-acetylation and thereby enhances LytA binding to peptidoglycan via the glycan-binding motif, which triggers the LytA-PtvBC-DimA signaling cascade to promote *hsdS$_A$* inversions toward *hsdS$_{A1}$*-OFF allelic configuration and a "starvation" methylome; in the nutrient-rich niches (e.g., inflamed upper airway, lungs and bloodstream), the bacterium would adopt a *hsdS$_{A1}$* allelic configuration and a "sufficiency" methylome; the "starvation" and "sufficiency" epigenetic states are manifested as the T and O colony phenotypes.

## Materials and methods

### Bacterial strains and cultivation

All the bacterial strains used in this study are summarized in S7 and S8 Tables. *S. pneumoniae* clinical isolate ST556 and its streptomycin-resistant derivative ST606 (ST556 *rpsL1*, containing a point mutation in ribosomal protein small subunit L) were used as the parental strains for mutant construction unless otherwise indicated [25]. *E. coli* strain DH5α for harboring specific plasmids and BL21(DE3) for producing recombinant proteins were bought from Solarbio company (Beijing, China). *E. coli* BTH101 was used as the reporter strain in the BATCH system [63]. Luria-Bertani (LB) broth was used for culturing *E. coli* strains. Pneumococci were cultured in a chemical-defined medium (CDM) with yeast extract (C + Y medium), tryptic soy broth (TSB), or on tryptic soy agar plate at 37°C with 5% $CO_2$ [28]. CDM was prepared according to

previous studies [62]. Before cultivation in CDM with different concentrations of glucose, pneumococci were incubated in C+Y broth to an optical density at 620 nm ($OD_{620}$) of 0.5, subsequently washed with PBS and resuspended in CDM to an initial $OD_{620}$ of 0.01. Appropriate antibiotics were added to the media when necessary.

## Chemicals and reagents

All commercial culture media were purchased from BD (NJ, USA). All the premixed or ingredients of chemicals were purchased from Sigma (Shanghai, China) unless otherwise described. All reagents and commercial kits for molecular biology procedures were obtained from New England Biolabs (Beijing, China) unless otherwise described.

## Bacterial mutagenesis

Pneumococcal mutants were constructed as described [28]. Markerless mutants were derived from streptomycin-resistant parental strains, including ST606 (556 *rpsL1*), TH6671 (P384 *rpsL1*), TH6675 (ST877 *rpsL1*), and TH6552 (the *hsdS~A1~*-fixed strain in ST556 *rpsL1* background) using JC1 (a modified Janus cassette) replacement method [28]. PgdA, Adr and LytA point mutants were established by *in situ* replacing JC1 sequence with the fusion PCR products of up- and down-stream sequences of target regions. In *pgdA*$^{D275N}$, the 823$^{rd}$ G of *pgdA* was changed to A based on a previous study [64]; in *adr*$^{S438A}$, the 1,312$^{th}$ T of *adr* gene was changed to A [30]; in *lytA*$^{E87A}$, the 260$^{th}$ A of *lytA* was changed to C; in *lytA*$^{H133A}$, the 397$^{th}$ C and 398$^{th}$ A of *lytA* were changed to G; in *lytA*$^{S33Q-Y41A}$, 97$^{th}$ T, 98$^{th}$ C, 121$^{st}$ T and 122$^{nd}$ A for *lytA* were changed to C, A, G, and C, respectively [35]. The deletion mutants of *myy0041*, *myy0606*, *myy0713*, *myy0734*, *myy1361*, *myy1406*, *myy1427*, *myy1585*, and *myy1950* were constructed by replacing their entire encoding regions with chloramphenicol resistance gene *cat* (amplified from the plasmid pIB166) [65]. The relevant plasmids, primers, and genetic manipulations are summarized in S9, S10 and S11 Tables, respectively.

## Microscopic quantification of O and T colonies

The opacity of pneumococcal colonies (colony phase) was observed after incubation on catalase-TSA under 37°C, 5% $CO_2$ for 17 h as described previously [28]. The number of O and T colonies in the central area on each plate (circling approximately 100 colonies) were quantified. The representative colonies of each strain on the catalase-TSA plate were photographed as the same time under a dissection microscope at magnification of 2×10 times [28].

## RNA sequencing

RNA-seq was performed by Novogene Bioinformatics Technology (Tianjin, China) as described [28]. Significant difference of transcripts between ST606 and its derivatives was defined by a cut-off value of fold change ≥1.5 and $P_{adj}$<0.05. Genes with less than 30 read counts were excluded.

## Quantitative real-time reverse transcriptase PCR

The relative proportion of *hsdS~A1~*-carrying bacteria in single populations of ST606 was determined by assessing the abundance of *hsdS~A1~* mRNA with quantitative real-time reverse transcriptase PCR (qRT-PCR) as described [28]. The relative proportions of bacteria carrying *hsdS~A1~*, *hsdS~A2~*, *hsdS~A3~*, *hsdS~A4~*, *hsdS~A5~*, and *hsdS~A6~* in individual populations were determined by quantifying the abundance of their respective mRNAs using qRT-PCR. Primers used for qRT-PCR are listed in S12 Table.

## Detection of *hsdS* gene configurations

The orientations of *hsdS* genes in the *cod* locus were determined by qPCR with the specific primer pairs (listed in S12 Table) targeting the three inverted repeats (IR1.1/1.2, IR2.1/2.2 and IR3.1/3.2) as described [26]. The relative abundance

of each IR in different directions is presented as $2^{-(\Delta CT)}$. And the ratio between the relative abundance of forward and reverse sequence of each IR was calculated and described in percentage. The sum of the relative abundance of forward and reverse sequence of each IR was defined as 100%.

## Genome sequencing

Genome sequence of the spontaneous mutant TH11857 (ST606 $hk11^{rev*}$) was determined by the next generation sequencing as described [66]. Genomic methylation was detected by single molecule real-time (SMRT) sequencing on PacBio RSII platform as described [28]. Genome sequencing and SMRT sequencing were performed by Novogene Bioinformatics Technology (Tianjin, China).

## *In vivo* co-immunoprecipitation

The *in vivo* co-immunoprecipitation (Co-IP) was performed to identify proteins that potentially interact with LytA and PtvB as described, with minor modifications [67]. The following strains expressing Strep-tagged proteins in various strain backgrounds were constructed in ST606 (TH16167, *Strep-lytA*; TH17335, *Strep-ptvB*) and $adr^{S438A}$ (TH16192, $adr^{S438A}$ *Strep-lytA*; TH17336, $adr^{S438A}$ *Strep-ptvB*) as described [28]. Strep-tag II was fused to its N-terminal of LytA and C-terminal of PtvB. These strains were cultured in TSB with 600 U/ml of catalase to an $OD_{620}$ of 0.6, and immediately cooled on ice. Bacteria were washed with buffer W (20 mM HEPES, pH 8.0, 100 mM NaCl) and subsequently treated with 1% formaldehyde to induce protein-protein cross-linking as described [68]. Tris-HCl (pH 8.0) was supplemented into bacterial suspension to the final concentration of 250 mM to terminate reaction, followed by wash using the buffer W. Bacteria were resuspended in pre-cooled lysis buffer (20 mM HEPES, pH 8.0, 100 mM NaCl, 1% Triton X-100) containing protease inhibitors, and was homogenized using the French Pressure Cell. The lysate was centrifuged at 4°C to remove cell debris. The supernatant was co-incubated with Strep-Tactin Sepharose 50% suspension (IBA, Germany); beads bound to target proteins were collected and washed by centrifugation and resuspension with buffer W. Bound proteins were eluted with buffer E (20 mM HEPES, pH 8.0, 100 mM NaCl, 5 mM desthiobiotin). The resulting proteins were detected by SDS-PAGE; protein bands were excised for liquid chromatography-tandem mass spectrometry (LC-MS/MS) analysis. The spectra from each run were searched against *S. pneumoniae* ST556 database using Proteome Discovery searching algorithm (v1.4) [69].

## Bacterial adenylate cyclase-based two-hybrid assay

Protein interaction was assessed using BATCH system based on the interaction-mediated reconstitution of two complementary adenylate cyclase fragments as described [29]. Specifically, *lytA* was cloned into pKT25 and pKNT25 to generate plasmids encoding LytA with T25 fused to the N-terminal (T25-LytA) or C-terminal (LytA-T25). *ptvB* (lacking transmembrane region) and *pcpA* (lacking signal peptide) were cloned into pUT18C and pUT18 to generate vectors expressing *ptvB* and *pcpA* with T18. To study pairwise interactions among PtvA, PtvB and PtvC, *ptvA* and *ptvB* were cloned into pKT25, while *ptvB*, *ptvC* and *ptvBC* (*ptvB* and *ptvC* are co-transcribed) were inserted into pUT18C to establish vectors encoding PtvB, PtvC, and PtvBC with T18 fused to the N-terminal. To test pairwise interactions between DimA and PtvA, PtvB, and PtvC, *ptvC* gene was cloned into pKT25 to generate plasmid encoding PtvC with T25 fused to the N-terminal (T25-PtvC), while *dimA* was cloned into pUT18C to generate vectors encoding *dimA* with T18 fused to the N-terminal (T18-DimA) and C-terminal (DimA-T18). The reverse tagged pair was constructed similarly by cloning *dimA* into pKT25 to generate T25-DimA. To test pairwise interaction between PsrA and DimA, the *psrA* gene was cloned into pKT25 and pKNT25 to generate plasmid encoding PsrA with T25 fused to the N-terminal (T25-PsrA) or C- terminal (PsrA-T25), respectively.

pKT25-*zip* and pUT18C-*zip* respectively encoding the T25- and T18-fused leucine zipper (35-aa-long, derived from a yeast transcriptional activator protein GCN4) were used as the positive control [63,70]. Reporter *E. coli* BTH101 strain was inoculated on the MacConkey/maltose plate containing ampicillin (100 μg/ml), kanamycin (100 μg/ml), and IPTG (0.5 mM)

for 4–8 days until the colonies of positive control became fuchsia. In addition to visual assessment, β-galactosidase activity of each reporter *E. coli* BTH101 strain was measured to quantify the functional complementation of T25 and T18 mediated by the interactions as described [70].

### Western blotting

LytA and pyruvate oxidase SpxB were detected by Western blotting using rabbit anti-LytA and anti-SpxB antisera as described previously [71]. The density of each protein band was digitized with the ImageJ software (ImageJ 1.47v; National Institutes of Health) on the basis of its chemiluminescence intensity level. The relative protein abundance of LytA was calculated by normalizing the protein band density of LytA to that of SpxB.

### Acetyl-CoA quantification

Acetyl-CoA was measured using Acetyl-CoA Content Assay Kit (Solarbio, China) according to the supplier's instructions. Bacteria were harvested, washed twice with ice-cold PBS, and sonicated. After centrifugation, cell lysates were analyzed in a 96-well plate.

### Vancomycin tolerance

Pneumococcal vancomycin tolerance was evaluated as described [38]. Briefly, bacteria were cultured to an $OD_{620}$ of 0.5 in THY medium. Culture aliquots were incubated in the presence or absence of 0.5 µg/ml vancomycin. Bacterial viability was measured by plating for CFUs at 3, 6, and 18 h post treatment.

### Quantification of NAM and NAM *O*-acetylation

Cell wall materials were extracted according to the previous study [15]. The extent of NAM *O*-acetylation was assessed as described [72]. To quantify NAM *O*-acetylation, cell wall extracts were treated with 0.2 M NaOH to saponify the *O*-acetyl group. The produced acetate was further derivatized and quantified using Dionex Ultimate 3000 UPLC system coupled to a TSQ Quantiva Ultra triple-quadrupole mass spectrometer (Thermo Fisher, CA) (equipped with a heated electrospray ionization probe in negative ion mode) as described previously [73]. Data analysis and quantitation were performed by the software Xcalibur 3.0.63 (Thermo Fisher, CA). NAM in pneumococcal cell wall was measured as described previously [74].

### Statistical analysis

All the original data were summarized in S13 Table and analyzed by GraphPad Prism. The ratio between O and T colonies and the proportion between forward and reverse IR-bound sequences in different pneumococcal strains were analyzed by two-sided Chi-square test, Yates' continuity corrected Chi-square test, or Fisher's exact test (by means). The difference of relative $hsdS_{A1}$ mRNA abundance, gene expression, and the relative activity of β-galactosidase were evaluated by two-tailed unpaired Student's *t* test. Bacterial CFU values in vancomycin tolerance experiment were analyzed by two-way ANOVA. Differences with a *P* value of < 0.05 (*), < 0.01 (**), < 0.001(***) or < 0.0001 (****) are defined as statistically significant.

## Supporting information

**S1 Fig. Colony morphology of *adr* and *pgdA* mutants constructed in serotype-6A (TH6671) and serotype-35B (TH6675) strains.** Red and blue arrowheads indicate the representative opaque (O) and transparent (T) colonies, respectively.
(TIF)

**S2 Fig. The *hsdS* gene configurations in the *adr* mutant of serotype-2 strain D39. (A)** The proportions of six $hsdS_A$ allelic variants in single populations of strain D39 or its $adr^{S438A}$ derivative were assessed by qRT-PCR using allele-specific primer sets. Data shown as mean ± s.d. of 3 replicates in a representative experiment. **(B)** The ratio of IR1-, IR2-, and IR3- bound sequences in different orientations in strain D39 or its $adr^{S438A}$ derivative are shown as in Fig 2E.
(TIF)

**S3 Fig. The impact of LytA on pneumococcal colony phase and *hsdS* inversions. (A to B)** The colony phenotypes (A) and relative abundance of the $hsdS_{A1}$ mRNA (B) of *lytA* mutant. **(C)** The abundance of LytA in ST606 (WT) deriva- tives. LytA was assessed by Western blotting using a rabbit antiserum (left panel). The relative protein abundance of LytA was calculated by normalization to the band density of the internal control pyruvate oxidase SpxB (right panel). **(D)** The Adr abundance in the whole protein lysates of ST606 and $adr^{S438A}$ strains is presented as the average of the peak area obtained from two biological repeats in a representative experiments. **(E)** Detection of interactions between LytA and its associated proteins by bacterial two-hybrid assay. The β-galactosidase activity is assessed and presented for each reporter strain. PC, positive control (pKT25-*zip* and pUT18C-*zip*), NC, negative control (empty vectors pKT25 and pUT18C). Significance between NC and experimental groups is presented.
(TIF)

**S4 Fig. Vancomycin tolerance of *ptvR* and *ptvR-lytA* double mutant in ST606 strain background.** Pneumococci were cultured to an $OD_{620}$ of 0.5 in THY medium before being incubated in the presence or absence of 0.5 µg/ml vanco- mycin under routine pneumococcal culture conditions. Bacterial survival was assessed by plating for CFU at various time points.
(TIF)

**S5 Fig. Assessment of physical interactions between PtvC and DimA. (A)** The genetic (upper panel) and protein (lower panel) features of the *ptv* locus. The *ptvR* gene encodes a negative regulator of this operon. The nucleotides between two adjacent genes are marked in base pairs (bp). The promoter and rho-independent transcription terminator are indicated by a black arrow and a hairpin. Lower panel depicts the predicted protein structure of PtvA, PtvB, and PtvC. The number of the amino acid (aa) at various regions are indicated. The transmembrane topology was predicted using TMHMM - 2.0 tool. **(B)** Detection of interactions between PtvC and DimA by bacterial two-hybrid assay. Colonies on the MacConkey/maltose plates (left panel) and β-galactosidase activity (right panel) are shown for each reporter strain. PC, positive control (pKT25-*zip* and pUT18C-*zip*), NC, negative control (empty vectors pKT25 and pUT18C). BC, blank control without plasmid. Significance between NC and the experimental group is presented.
(TIF)

**S6 Fig. Impact of carbon metabolism on colony opacity and expression of *adr*. (A)** Representative colonies of *acoB* and *pfl* mutants. Top panel indicates the organization of genes encoding PDHC. Colonies indicated by red and blue arrow- heads represent O and T colonies, respectively. **(B)** The transcription of *adr* in pneumococci cultured in CDM with different concentrations of glucose. The mRNA of *adr* was detected by qRT-PCR and normalized to that of the internal control *era*.
(TIF)

**S1 Table. Methylated DNA motif specified by Spn556I/III MTase.**
(DOCX)

**S2 Table. SMRT sequencing data of *pgdA* and *adr* mutants.**
(DOCX)

**S3 Table. LytA-associated proteins changed in the Adr-inactivated mutant.**
(DOCX)

**S4 Table. PtvB-associated proteins changed in the Adr-inactivated mutant.**
(DOCX)

**S5 Table. Differentially expressed genes between *adr* mutant and WT strain in RNA sequencing.**
(XLSX)

**S6 Table. Differentially expressed genes between *adr* and *adr-lytA* mutants in RNA sequencing.**
(XLSX)

**S7 Table. Information of pneumococcal strains used in this study.**
(DOCX)

**S8 Table. Information of *E. coli* strains used in this study.**
(DOCX)

**S9 Table. Information of plasmids used in this study.**
(DOCX)

**S10 Table. Primers used for mutant construction in this study.**
(DOCX)

**S11 Table. Construction of bacterial mutants in this study.**
(DOCX)

**S12 Table. Primers used for qRT-PCR and qPCR in this study.**
(DOCX)

**S13 Table. The original data for statistical analysis in this study.**
(XLSX)

## Acknowledgments

We thank the National Protein Science Facility in Tsinghua University for assistance in protein mass spectrometry (Center for Proteomics) and NAM *O*-acetylation analysis (Metabolomics and Lipidomics Center).

## Author contributions

**Conceptualization:** Xiu-Yuan Li, Juanjuan Wang, Jing-Ren Zhang.

**Data curation:** Xiu-Yuan Li, Ping He, Shaomeng Wang, Juanjuan Wang.

**Formal analysis:** Xiu-Yuan Li, Ping He, Shaomeng Wang, Zhixing Feng, Juanjuan Wang.

**Funding acquisition:** Juanjuan Wang, Jing-Ren Zhang.

**Investigation:** Xiu-Yuan Li, Shaomeng Wang, Juanjuan Wang, Jing-Ren Zhang.

**Methodology:** Xiu-Yuan Li, Yusong Wang, Dingfei Yan, Xiaohui Liu, Haiteng Deng.

**Project administration:** Jing-Ren Zhang.

**Resources:** Xiaohui Liu, Haiteng Deng, Jing-Ren Zhang.

**Software:** Xiu-Yuan Li, Ping He, Yusong Wang, Dingfei Yan, Zhixing Feng.

**Supervision:** Juanjuan Wang, Jing-Ren Zhang.

**Validation:** Xiu-Yuan Li, Juanjuan Wang, Jing-Ren Zhang.

**Visualization:** Xiu-Yuan Li, Juanjuan Wang, Jing-Ren Zhang.

**Writing – original draft:** Xiu-Yuan Li.

**Writing – review & editing:** Xiu-Yuan Li, Juanjuan Wang, Jing-Ren Zhang.

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
