## [Decision Letter · Decision Letter 0]

Dear Dr. Zhang,

Thank you very much for submitting your manuscript "*Streptococcus pneumoniae*  synchronizes the states of cell wall peptidoglycan acetylation and genome methylation by programmed DNA inversions" for consideration at PLOS Pathogens. As with all papers reviewed by the journal, your manuscript was reviewed by members of the editorial board and by several independent reviewers. In light of the reviews (below this email), we would like to invite the resubmission of a significantly-revised version that takes into account the reviewers' comments.

Your manuscript was read in detail by three experts in the field. Although the reviewers found your manuscript to be interesting, important and a significant addition to the field, they identified some concerns that need to be addressed in a revised version to fully validate all the claims put forth.

We cannot make any decision about publication until we have seen the revised manuscript and your response to the reviewers' comments. Your revised manuscript is also likely to be sent to reviewers for further evaluation.

Sincerely,

Anders P Hakansson, Ph.D.

Academic Editor

PLOS Pathogens

Helena Boshoff

Section Editor

PLOS Pathogens

Michael Malim

Editor-in-Chief

PLOS Pathogens

orcid.org/0000-0002-7699-2064

Your manuscript has been in detail by three experts in the field. Although the reviewers found your manuscript to be interesting, important and a significant addition to the field, they identified some concerns that need to be addressed.

Reviewer's Responses to Questions

**Part I - Summary**

Reviewer #1: This submission analyzes pneumococcal factors affecting DNA inversions controlling differences in genome methylation. The findings support a novel role for differences in peptidoglycan O-acetylation. A contribution of the amidase, LytA, is also demonstrated in these events, although it remains unclear how bacterial factors shown to associate with LytA affect DNA inversions. It is also postulated that O-acetylation occurs in response to differences in the availability of acetyl-CoA and, thus, the metabolic state of the organism. This last part of this otherwise detailed and extensive manuscript is particularly interesting but, in my opinion, requires further substantiation. Below are some further comments that could improve the manuscript.

Reviewer #2: The observation that colony morphology of S. pneumoniae varies between transparent (T) and opaque (O) and correlates with many important phenotypes, including virulence and site of colonization of the mammalian host, was made decades ago, but the molecular basis for this variation has remained unknown. This manuscript by Zhang provides new and important insights into the regulation of colony morphology, identifying O-acetylation of peptidoglycan (PG) as a key signal. PG acetylation, in turn, is likely to be influenced by the intracellular concentration of acetyl-CoA, thus linking the metabolic state (and the environmental nutrient availability) to this phase variation. This work builds on their previous discovery of a locus that is subject to variation by site-specific recombination and controls DNA methylation and thus the epigenetic state of the bacterium.

The manuscript is well-written and the work extremely comprehensive. I would require no new experiments. The issues I raise below are to enhance clarity, to point out typos, or to simply raise points that the authors can, if they think useful, address in some non-experimental manner, e.g., in the Discussion.

Reviewer #3: This is a very interesting, ambitious paper that presents some potentially very important findings. The authors provide convincing data for a genetic link between O-acetylation of PG MurNAc and the formation of transparent (T) or opaque (O) colonies in certain pneumococcal serotype strains. The switch between O and T colony morphologies has previously been shown to be correlated with phase variation of an invertible sequences in the hsdS gene encoding the DNA-recognition protein of the restriction-modification system that methylates different sequences in chromosome. Previous work showed that the T/O and hsdS epigenetic switch is controlled by four two-component-regulatory systems, especially HK/RR TS11. This study considerably extends these previous observations by showing that the function of the Adr O-acetylase enzyme also impacts this epigenetic switch. Some of the data in this regard are compelling, such as the demonstration that inactivation of Adr prevents the HsdS-A1 orientation that is correlated with the O state colony formation. Moreover, the paper goes on to show that this effect of Adr inactivation is somehow linked to the functions of LytA PG autolysin and some other proteins that directly or indirectly interact with LytA. Finally, the authors relate the phase variation switch to the function enzymes involved in synthesizing Acetyl-CoA, which is required for the Adr-catalyzed PG O-acetylation.

In some ways, this is an outstanding study that presents some novel and highly interesting results. The presentation is generally clear and compelling. On the other hand, there are some issues about what state the bacteria are in physiologically and how the proposed mechanism could function. Notably, the study does not loop in other studies on the activation of LytA to the interpretation of the findings. There are also several places were modification and clarification are required. Suggested changes are listed below, with major issues marked by two asterisks (**).

**1. Line2 25, 375, 376, and throughout. The term O-de-acetylation is misleading and incorrect. It should be changed and throughout, especially in the Discussion. O-de-acetylation implies an enzymatic process, such as the GlcNAc de-acetylation carried out by PdgA. No O-de-acetylase enzyme has been reported in pneumococcus. Please change “O-de-acetylation” to “lack of O6-acetylation,” “absence of O6-acetylation” or “decreased O6-acetylation”.

2. Lines 25-27. Typo (remove “of”). Also, please consider revising this sentence. It seems backward in that acetyl-Co-A is at the end of the sentence. Also, “physiological stages” is ambiguous.

3. Line 49. Delete first “the”.

**4. Indicate that O6 acetylation and attachment of WTAs are mutually exclusive. Please see comments later about LytA activation by WTA binding.

5. **Line 67 after [16]. Please rearrange Fig. 1 so that Fig. 1C is Fig. 1A, and include “(see Fig. 1A)” after “[16]”. It will be hard for readers to visualize this part without a diagram.

Line 69. Ambiguous sentence. Consider changing to “and protects against autolysis”.

6. Line 75. Typo.

7. **Line 75. After [23,24] add “(see Fig. 2A)”. It will be hard for readers to visualize this part without a diagram.

8. Lines 91-95. Awkward sentence. Please rephrase.

9. **Line 102. The type in all of the figures is much too small to read easily, especially the many bar diagrams. There is room to use larger size type in all of the figures and this should be done for clarity. Also, the reproduction of the larger figures was blurry and pixelated. Please these problems.

10. **Line 103 and everywhere. The term “complementation” is not used properly in a genetic sense. Complementation means retaining a mutation in the chromosome and providing a separate wild-type copy of a gene ectopically. Complementation rules out polarity and other aberrations caused by the mutation and assigns function to the gene being interrogated. If I understand the strain table correctly, these “rev” strains contain repair of mutations back to WT. That is, the WT copy of the gene has replaced the mutant copy in the chromosome. The term “complementation” needs to be changed throughout. Were actual genetic complementation experiments performed? Some of the broader conclusions are weakened somewhat by lack of proper complementation experiments.

11. **Line 103 and everywhere. The fact that point mutations were also constructed in most genes that were deleted and then repaired does tend to argue against polarity and downstream effects, although the way the strains are presented in the supplemental tables makes it somewhat difficult to tell where when antibiotic markers are present. Where these mutations markerless? In addition, point mutations often have the same phenotypes as the deletions, and it is not possible to tell without western blotting whether the mutants with amino acids were expressed or degraded. Was protein expression confirmed by western blotting in critical experiments?

12. **Line 112. It is highly informative that similar results were detected in two other serotype strains. However, what happened in the serotype 2 strains in which most previous work on LytA and Adr was performed. This should be easy to check, because the authors have shown the rearrangements of hsdS in the D39 strain.

13. Line 128. Please consider changing “modulates” to “favors”.

14. Line 162. Please consider changing “compensation” to “restoration” or “repair”.

15. **Line 147. PgdA clearly modulates colony opacity, but not through the hsdS mechanism. Then PgdA is completely dropped from the study. How is PgdA doing this and does it impact the conclusions about Adr modifications? Please explain. Related to this question, why did A1 drop significantly in the pgdA mutant (Fig. 3B, column 2),

16. Line 174. Remove “have” in sentence.

17. **Line 196 and throughout. It has been shown that in cells untreated for antibiotics LytA is active primarily in stationary-phase cells. Cells in T/O colonies are likely in deep stationary phase. However, it is not clear in the other experiments what growth phase the cells were taken in. The one growth curve is in Figure 7D and the Methods suggest that cells were taken between 0.4 and 0.6 OD620, which appears to be early stationary phase. What happens in early to middle exponential phase? Does the epigenetic switching still occur when LytA is not active?

18. **Line 196, Fig. 7, and throughout. Along the same lines, some key papers about LytA function, such as Flores-Kim et al. eLife 8:e44912 2019 are not cited or considered in the model. Moreover, assuming that the model in Fig. 7F is correct, why wouldn’t the constitutive PG cell wall damage by LytA be lethal in the O-phase cells. Also, how accurate is the idea that colonizing pneumococci are starving during colonization (e.g., line 493)? Work by Samantha King and other others have shown that pneumococcal cells produce a variety of glycosyl transferases that cleave sugars off of the very abundant glycoproteins on epithelial cells. Please clarify this matter more.

19. **Line 222. Were western blots performed to determine whether these domains were actually expressed and at what relative levels?

20. **Line 248. The list in Table S2 contains some cytoplasmic proteins. Are you implying that these proteins bind to LytA before it is secreted? Are there any corroborative data (e.g., B2H assays) that PtvB binds to LytA?

21. **Line 253. Is the involvement of PtvB in vancomycin tolerance dependent on LytA? This phenotype would support the model of an interaction.

22. Line 268. Awkward sentence; too many WTs. Please rephrase.

23. **Line 279. As with the other proteomics data, the putative interactors seem to be all over the place. What does this mean?

24. **Line 287. Why is the methyltransferase mutant viable? Without methylation, why doesn’t the restriction enzyme kill the cell? Please explain.

25. Line 295. The dimA nomenclature is used in line 285, before naming it in line 295. Please move the naming up to around line 285.

26. Line 303. Typo.

27. **Line 308. This binding of DimA to PtvC seems marginal and not significant as stated. Did the reverse pair, show this binding more strongly? AS it stands, this is not a strong conclusion.

28. Line 316. Please include a protein interaction diagram to summarize the interactions and their relative strengths.

29. Line 328 and Fig. 7B. Do you mean “rate” as stated in the figure or amount? Amount seems more appropriate.

30. **Fig. 7D. The changes in the orientation of hsdS seem very modest. Are these very most changes sufficient to cause biological effects?

31. **Fig 7E. At what stage in growth are the glucose replete cultures. This growth curve does not seem correct. Is it pre-exponential? Please re-graph on a log scale.

32. **Line 351. Does the cellular amount of Adr protein change under the different growth conditions? Could a change in Adr protein amount contribute to these phenotypes?

33. Line 373. Change “is” to “may”.

34. **Line 385-401. Please see comments about LytA regulation and growth phase above. In addition, are any interactors of PsrA known that could tie together the results in this paper?

35. Lines 402-409. The comparison to an anti-sigma factor mechanism is a bit confusing and could be left out.

36. **Line 459. Are any of the proteins reported here included in the regulons of these TCSs? Please comment on this, because it could provide a link.

37. Line 482. Change “represents a major breakthrough” to “is a potential breakthrough” since the mechanism and growth phase dependence is far from clear.

38. Line, 500. Add: “,respectively.”

39. **Does O6-acetylation significantly reduce WTA amount? Since TA binding is required for LytA activity, could this contribute to the block in LytA activity postulated in Fig. 7F right. Again, please provide context for the current interpretations based on previous functional studies of LytA, such as in Flores-Kin et al, 2019, listed above.

40. **As a general comment, as presented it is difficult to match the strains used in the experiments to the many bar graphs. The strain numbers from Table S4 should be listed in the figure legends.

**Part II – Major Issues: Key Experiments Required for Acceptance**

Reviewer #1: 1) The narrative implies that pneumococci have evolved to allow for survival during invasive infection. Since all forms of invasive infection are a dead end for this organism, this is difficult to accept as stated in the Introduction and Discussion sections. On the other hand, it seems more plausible that the organism adapts to different nutrient conditions, although these must be during colonization or transmission – the productive outcomes for the organism.

2) The important data in Figure 3 would be more convincing if a revertant of the key strain HsdSa1adrS438A was included.

3) The transition to the consideration of LytA (and other choline binding proteins) is difficult to follow (beginning on lines 202-3). Please explain this in more detail. Of note, differences in LytA binding to O and T forms was previously reported (PMID:8675333) and should be discussed.

4) My major criticism is that differences in acetyl-CoA are described as an important aspect of regulation but never actually shown. Genes in the PDHC pathway (Fig. 7A) could have other effects and the mechanism described could be due to indirect consequences of manipulating this pathway. It would be more convincing to measure and directly manipulate acetyl-CoA levels to justify this key claim.

5) The final paragraph of the Results is not very convincing/impressive. Growth phase dependence only leads to a minor, albeit statistically significant, change in hsdSA1 abundance from 44.7 to 32.3% (Fig. 7D. Likewise the effect of glucose concentration (Fig. 7E) is small at best.

6) Doesn’t deletion of lytA alone affect colony morphology? …..Independently of gene conversions? Can lytA mutants truly be compared (Fig4C) to lytA+?

Reviewer #2: (No Response)

Reviewer #3: See points marked by ** in review above.

**Part III – Minor Issues: Editorial and Data Presentation Modifications**

Reviewer #1: Line 43: surface rather than faces

Line 105. The transition here is rough. Suggest changing to - Whole genome sequencing of hk11rev-N revealed……

Line 119: Change to: replacement of serine at position 438 with alanine

Line 132: Change The previous studies to: Previous studies

Fig 7F. Please add a ? where the dashed lines are shown. As noted above, I would delete nasal colonization and invasive infection labels as this is not shown.

Reviewer #2: 1. Have the authors ever compared the % of PG O-acetylation of wild type:

a. In different growth phases or media?

b. With any of their mutants, e.g., lytA mutant?

2. Similarly, have the authors measured intracellular actyl CoA in different growth phases or media and correlated with PG O-acetylation?

3. Is LytA expression or activity altered depending on growth phases or media? Biofilm vs. non-biofilm growth? If so, there are several downstream predictions from the authors’ model that could be tested.

4. Fig 1a. The naming of hkkrev-N and hkkrev strains is confusing. The “N” signifies the original strain that carries a mutation in the 5’ noncoding region of pdgA mutation, but the terminology seems non-intuitive, and the information is missing from the figure legend. Perhaps just an asterisk with note in legend that this indicates a spontaneous mutation in the non-coding region of pdgA.

5. Line 34: In abstract, the meaning of "cross-membrane functional linkage" is unclear.

6. LIne 60. For clarity, I would mention "cell wall teichoic acids, which contain choline"

7. Line 75. "determinant"

8. Line 66. “subjected”

9. A schematic figure illustrating hsd locus and inversion substrates would be useful.

10. Line 85. regulated by four of the 13 ...

11. Line 86. Mutation of any of these four TCS led to ...

12. Line 281. Upon first mention of dimA/myy1025, not that myy1025 is termed dimA based on evidence presented below.

13. Line 282. Unclear sentence. Consider “phenocopied the ptvB mutant in terms of …”

14. Line 287. Not clear why the authors make this statement—explain further or delete.

15. Fig. 6H. The authors should comment on the much more subtle phenotype of the ptvC mutant compared to the ptvB mutant

16. Line 328. Suggest softening by replacing “required for” with “utilized in the production of”.

17. Fig. 7 model and Line 394. A representation of the hsdS-A1-independent pathway to colony opacity should be added to the schematic. This might simply be some indication that this pathway is dependent on LytA catalytic activity, not simply PG binding, and perhaps an arrow with some question marks in the cytoplasmic space.

18. Line 444. The authors should also cite the extensive investigation into the role of Xis in directionality of lambda site-specific recombination (PMID: 25114241).

19. Line 489. The authors should consider “via the extracellular regulatory circuit that involves PG O-acetylation”.

Reviewer #3: See review above.

PLOS authors have the option to publish the peer review history of their article (what does this mean? ). If published, this will include your full peer review and any attached files.

**Do you want your identity to be public for this peer review?** For information about this choice, including consent withdrawal, please see our Privacy Policy .

Reviewer #1: No

Reviewer #2: No

Reviewer #3: No
---

## [Decision Letter · Decision Letter 1]

PPATHOGENS-D-24-01692R1

*Streptococcus pneumoniae*  synchronizes the states of cell wall peptidoglycan acetylation and genome methylation by programmed DNA inversions

PLOS Pathogens

Dear Dr. Zhang,

Thank you for submitting your revised manuscript to PLOS Pathogens. After careful consideration, we feel that there are minor aspects of the manuscript that still need to be revised prior to acceptance of your manuscript. Therefore, we invite you to submit a revised version of the manuscript that addresses the points raised during the review process.

Please submit your revised manuscript within 30 days Jul 27 2025 11:59PM. If you will need more time than this to complete your revisions, please reply to this message or contact the journal office at plospathogens@plos.org. Please include the following items when submitting your revised manuscript:

We look forward to receiving your revised manuscript.

Kind regards,

Anders P Hakansson, Ph.D.

Academic Editor

PLOS Pathogens

Helena Boshoff

Section Editor

PLOS Pathogens

Sumita Bhaduri-McIntosh

Editor-in-Chief

PLOS Pathogens

orcid.org/0000-0003-2946-9497

Michael Malim

Editor-in-Chief

PLOS Pathogens

orcid.org/0000-0002-7699-2064

**Additional Editor Comments:**

Your revised manuscript has now been re-evaluated by the reviewers and all three reviewers concur that the additional data and revision of your manuscript has significantly improved the study and now provides a high-quality study on a very important topic. I would, however, prior to accepting your manuscript for publication, ask that you address the concerns of Reviewer 1 regarding the interpretation of the direct role of Acetyl-CoA in the epigenetic regulation.

**Journal Requirements:**

Please amend your detailed Financial Disclosure statement. This is published with the article. It must therefore be completed in full sentences and contain the exact wording you wish to be published. State the initials, alongside each funding source, of each author to receive each grant. For example: "This work was supported by the National Institutes of Health (####### to AM; ###### to CJ) and the National Science Foundation (###### to AM).".

**Reviewers' Comments:**

Reviewer's Responses to Questions

**Part I - Summary**

Reviewer #1: This is an excellent and thorough study of the relationship between cell wall modification and genome methylation. That said, this reviewer still finds the link to AcCoA levels and nutrient availability in Figure 8 to be greatly overstated in the revised manuscript.

Reviewer #2: Strong study made stronger by revision.

Reviewer #3: The authors did an excellent job in responding to all of the previous reviewer comments by adding data and revising parts of the manuscript. This is an interesting, important contribution to this area. The manuscript contains a large amount of high-quality data leading to solid conclusions and is presented well. I have no additional comments.

**Part II – Major Issues: Key Experiments Required for Acceptance**

Reviewer #1: While Figure 8 establishes an effect of growth phase on methylation, a direct relationship to AcCoA levels is still not demonstrated. For example, Fig. 8L shows a significant decease in AcCoA levels between 2.5 and 1 g/L of glucose but the effect on hsdSa1 abundance in Fig. 8M is observed at a glucose concentration of between 20 and 10 g/L - a ten fold difference. This does not correlate. In my view, the claim in lines 41-45 (abstract) and corresponding section of the results and discussion (and model in Fig 9) about nutrient control needs to be substantially revised to remove any direct relationship to AcCoA.

Reviewer #2: None

Reviewer #3: None

**Part III – Minor Issues: Editorial and Data Presentation Modifications**

Reviewer #1: 1) The last 'summarizing' sentence in the abstract is confusing and should be simplified.

2) The excessive use of strongly, dramatically, unequivocally, and striking are unnecessary and distract from the narrative. The strong data speaks for itself.

3) line 122, overhauls? change word.

Reviewer #2: None

Reviewer #3: None

PLOS authors have the option to publish the peer review history of their article (what does this mean? ). If published, this will include your full peer review and any attached files.

**Do you want your identity to be public for this peer review?** For information about this choice, including consent withdrawal, please see our Privacy Policy .

Reviewer #1: No

Reviewer #2: No

Reviewer #3: No

**Figure resubmission:**
---

## [Editor Report · Decision Letter 2]

Dear Dr. Zhang,

We are pleased to inform you that your manuscript '*Streptococcus pneumoniae*  synchronizes the states of cell wall peptidoglycan acetylation and genome methylation by programmed DNA inversions' has been provisionally accepted for publication in PLOS Pathogens.

Best regards,

Anders P Hakansson, Ph.D.

Academic Editor

PLOS Pathogens

Helena Boshoff

Section Editor

PLOS Pathogens

Sumita Bhaduri-McIntosh

Editor-in-Chief

PLOS Pathogens

orcid.org/0000-0003-2946-9497

Michael Malim

Editor-in-Chief

PLOS Pathogens

orcid.org/0000-0002-7699-2064

The authors have addressed the concerns of Reviewer !.
---

## [Editor Report · Acceptance letter]

Dear Dr. Zhang,

We are delighted to inform you that your manuscript, "*Streptococcus pneumoniae*  synchronizes the states of cell wall peptidoglycan acetylation and genome methylation by programmed DNA inversions," has been formally accepted for publication in PLOS Pathogens.

Best regards,

Sumita Bhaduri-McIntosh

Editor-in-Chief

PLOS Pathogens

orcid.org/0000-0003-2946-9497

Michael Malim

Editor-in-Chief

PLOS Pathogens

orcid.org/0000-0002-7699-2064